# Solving hidden monotone variational inequalities with surrogate losses

**Ryan D'Orazio, Danilo Vucetic & Zichu Liu**
Mila Québec AI Institute, Université de Montréal
Montréal, QC, Canada
`{ryan.dorazio, danilo.vucetic, zichu.liu}@mila.quebec`

**Junhyung Lyle Kim**
Department of Computer Science, Rice University
Houston, TX, USA
`jlylekim@rice.edu`

**Ioannis Mitliagkas & Gauthier Gidel**
Mila Québec AI Institute, Université de Montréal, CIFAR AI Chair
Montréal, QC, Canada
`{ioannis, gidelgau}@mila.quebec`

## Abstract

Deep learning has proven to be effective in a wide variety of loss minimization problems. However, many applications of interest, like minimizing projected Bellman error and min-max optimization, cannot be modelled as minimizing a scalar loss function but instead correspond to solving a variational inequality (VI) problem. This difference in setting has caused many practical challenges as naive gradient-based approaches from supervised learning tend to diverge and cycle in the VI case. In this work, we propose a principled surrogate-based approach compatible with deep learning to solve VIs. We show that our surrogate-based approach has three main benefits: (1) under assumptions that are realistic in practice (when hidden monotone structure is present, interpolation, and sufficient optimization of the surrogates), it guarantees convergence, (2) it provides a unifying perspective of existing methods, and (3) is amenable to existing deep learning optimizers like ADAM. Experimentally, we demonstrate our surrogate-based approach is effective in min-max optimization and minimizing projected Bellman error. Furthermore, in the deep reinforcement learning case, we propose a novel variant of TD(0) which is more compute and sample efficient.

## 1 Introduction

Most machine learning approaches learn from data by minimizing a loss function with respect to model parameters. Despite the lack of global convergence guarantees in deep learning, losses can often still be minimized with an appropriately tuned first-order adaptive method such as ADAM (Kingma, 2014). Unfortunately, outside of scalar loss minimization, deep learning becomes more challenging: the dynamics of variational inequality (VI) problems (*e.g.*, min-max) are often plagued with rotations and posses no efficient stationary point guarantees (Daskalakis et al., 2021). Thus, the additional challenges posed by VI problems do not allow one to easily plug in existing techniques from deep learning.

We propose a practical and provably convergent algorithm for solving VI problems in deep learning. Importantly, our approach is compatible with any black-box optimizer, including ADAM and its variants (Kingma, 2014), so long as they are effective at descending a scalar loss. Our approach guarantees convergence by leveraging the hidden structure commonly found in machine learning.

More precisely, many applications admit a hidden structure corresponding to the following problem:

$$\text{find } z_* \text{ such that } \langle F(z_*), z - z_* \rangle \geq 0 \quad \forall z \in \mathcal{Z} = \text{cl}\{g(\theta) : \theta \in \mathbb{R}^d\}, \tag{1}$$

where the mapping $g : \mathbb{R}^d \to \mathcal{Z} \subseteq \mathbb{R}^n$ maps model parameters to model outputs, and the set $\mathcal{Z}$ encodes the closure of the set of realizable outputs from the chosen model. Since $F$ is defined over model outputs, it will often be structured, e.g., monotone and Lipschitz.

In order to leverage the hidden structure in VI problems, we employ surrogate losses, which have been studied in the scalar minimization case by Johnson & Zhang (2020) and Vaswani et al. (2021), among others. The surrogate loss approach has been shown to be scalable with deep learning and has been used in modern reinforcement learning policy gradient methods (Schulman et al., 2015; 2017; Abdolmaleki et al., 2018). Moreover, these surrogate loss approaches have been shown to improve data efficiency in cases where evaluating $F$ is expensive like interacting with a simulator in reinforcement learning (Vaswani et al., 2021; Lavington et al., 2023).

Our approach reduces the VI in (1) to the approximate minimization of a sequence of surrogate losses $\{\ell_t\}_{t \in \mathbb{N}}$, which are then used to produce a sequence of parameters $\{\theta_t\}_{t \in \mathbb{N}}$. To ensure convergence we propose a new $\alpha$-descent condition on $\ell_t$, allowing for a dynamic inner-loop that makes no assumption on how the surrogate losses are minimized, thereby allowing for any deep-learning optimizer to minimize the scalar loss $\ell_t$. With our $\alpha$-descent condition we provide convergence guarantees to a solution in the space of model predictions $\{z_t = g(\theta_t)\}_{t \in \mathbb{N}} \to z_*$ for a sufficiently small $\alpha$. Our general method is summarized in Algorithm 1.

---

**Algorithm 1:** $\alpha$-descent on surrogate

---

**Input:** Outer loop interactions $T$, initial parameters $\theta_1 \in \mathbb{R}^d$, step size $\eta$ for surrogate loss,
$\quad$ $\alpha \in (0, 1)$, optimizer update $\mathcal{A} : \mathcal{L} \times \mathbb{R}^d \to \mathbb{R}^d$.
**for** $t = 1 \leftarrow$ **to** $T$ **do**
$\quad$ Compute VI operator: $F(g(\theta_t))$
$\quad$ Set surrogate loss: $\ell_t(\theta) = \frac{1}{2}\|g(\theta) - (g(\theta_t) - \eta F(g(\theta_t)))\|^2$
$\quad$ $\theta_s \leftarrow \theta_t$
$\quad$ **while** $\ell_t(\theta_s) - \ell_t^* > \alpha^2(\ell_t(\theta_t) - \ell_t^*)$ **do**
$\quad\quad$ Update parameters with optimizer: $\theta_s \leftarrow \mathcal{A}(\ell_t, \theta_s)$
$\quad$ $\theta_{t+1} \leftarrow \theta_s$
**return** $\theta_{T+1}$

---

In summary, we propose a novel algorithm to solve VI problems by exploiting the hidden structure of common loss functions. We demonstrate the efficacy of our method with a variety of experiments, which themselves make further connections and innovations on prior work. Our specific contributions are as follow:

- We provide the **first extension of surrogate losses to VI problems**. We also show a clear separation in difficulty of using surrogate methods in VI problems when compared to scalar minimization; specifically, we show that divergence is possible in a strongly monotone VI problem where in contrast convergence is guaranteed in the non-convex scalar minimization case.

- We propose the $\alpha$-**descent condition with convergence guarantees**. This condition allows for global convergence while avoiding common assumptions such as enforcing errors to be summable or globally upper-bounded.

- **Unifying perspective of pre-conditioning methods**. With our surrogate loss approach, we unify existing pre-conditioning methods (Bertsekas, 2009; Mladenovic et al., 2022; Sakos et al., 2024) by showing they are equivalent to using the Gauss-Newton method as the optimizer $\mathcal{A}$ in Algorithm 1 to minimize the surrogate losses. We demonstrate the value of this new perspective by providing natural extensions of their methods with better empirical robustness.

- **Experimental results and new TD variants**. We demonstrate the performance and versatility of surrogate loss-based optimization in a variety of VI problems. In Sections 5.1 and 5.2 we complete experiments in min-max optimization and value prediction tasks, respectively. Importantly, we propose a new data-efficient variant to TD(0) which significantly outperforms prior approaches.

## 2 BACKGROUND AND RELATED WORK

**Notation.** We use $\langle x, y \rangle = \sum_{i=1}^{n} x^i y^i$ to denote the standard inner product over $\mathbb{R}^n$ and $\|x\| = \sqrt{\langle x, x \rangle}$ to be the Euclidean norm. We write $\|x\|_\Xi^2$ to mean $\langle x, \Xi x \rangle$, and $\|x\|_\Xi$ is a norm if and only

if $\Xi$ is positive definite. For a set $\mathcal{X}$ we denote $\text{cl}\,\mathcal{X}$ its closure and $\text{ri}\,\mathcal{X}$ its relative interior. For a given set, which will be clear from context, we denote $\Pi(x)$ as the Euclidean projection of $x$ onto the set and similarly $\Pi_\Xi(x)$ the projection with respect to $\|x\|_\Xi$. We use $\text{Id}$ to denote the identity matrix. A matrix $A$ has lower and upper bounded singular values if there exists $\sigma_{\min}, \sigma_{\max} \in (0, \infty)$ such that for any $x$ we have $\sigma_{\min}^2 \|x\|^2 \le \langle x, A^\top A x \rangle \le \sigma_{\max}^2 \|x\|^2$. If a matrix $A$ is invertible we write $A^{-1}$ otherwise we denote the pseudo-inverse as $A^\dagger$. For a given function $g : \mathbb{R}^d \to \mathbb{R}^n$ we write $Dg(\theta)$ as its Jacobian evaluated at $\theta$. We say $Dg$ has uniformly lower and upper bounded singular values if there is a constant upper and lower bound to the singular values of $Dg(\theta)$ for all $\theta \in \mathbb{R}^d$.

**Surrogate Loss Background.** In the scalar minimization case such as supervised learning, a non-convex loss function of the form $f(g(\theta))$ is minimized where the non-convexity is due to the model parametrization as represented by the model predictions: $g(\theta) = \begin{bmatrix} g^1(\theta), \cdots, g^n(\theta) \end{bmatrix}^\top \in \mathbb{R}^n$. In supervised learning, each prediction is $z^i = g^i(\theta) = h(x_i, \theta)$, for some feature vector $x_i$ and fixed model architecture $h$. Despite the non-convexity in parameter space $\theta \in \mathbb{R}^d$ due to $g$, the loss function is often convex and smooth with respect to the closure of the predictions $\mathcal{Z} = \text{cl}\{g(\theta) : \theta \in \mathbb{R}^d\}$.[1] The optimization problem can then be reframed as a constrained optimization problem,

$$\min_\theta f(g(\theta)) = \min_{z \in \mathcal{Z}} f(z). \tag{2}$$

If $\mathcal{Z}$ is convex then projected gradient descent $z_{t+1} = \Pi(z_t - \eta \nabla f(z_t))$ is guaranteed to converge (Beck, 2017). However, the projection $\Pi$ is expensive since it is with respect to the set $\mathcal{Z} \subseteq \mathbb{R}^n$. In general, the model-dependent constraint $\mathcal{Z}$ may not be convex. Yet, this assumption is essential since we require convergence of the projected gradient method; we leave relaxing this assumption for future work. Nevertheless, this assumption is satisfied in two important extreme cases, when a model is linear or with large capacity neural networks that can interpolate any dataset e.g. $\mathcal{Z} = \mathbb{R}^n$ (Zhang et al., 2017).

Beyond supervised learning, similar hidden structure exists in losses within machine learning such as: generative models and min max optimization (Gidel et al., 2021), robust reinforcement learning (RL) (Pinto et al., 2017), and minimizing projected Bellman error (Bertsekas, 2009). However, in these applications the problem cannot be written as minimizing a loss and we must instead consider the VI problem (1). For example, in the min-max case, the min and max players' strategies may be given by two separate networks $h^1(\theta^1), h^2(\theta^2)$, respectively, with the following objective:

$$\min_{\theta^1} \max_{\theta^2} f(h^1(\theta^1), h^2(\theta^2)), \tag{3}$$

where $f$ is convex-concave. Similar to (2) we can rewrite the problem in parameter space as a constrained problem with respect to model predictions but instead within the VI (1); where $\theta = (\theta^1, \theta^2)$, and $g(\theta) = (h^1(\theta^1), h^2(\theta^2))$, with operator $F(z) = F(g(\theta)) = [\nabla_{z^1} f(z^1, z^2), -\nabla_{z^2} f(z^1, z^2)]^\top$ where $z^1 = h^1(\theta^1)$ and $z^2 = h^2(\theta^2)$. The solution with respect to model outputs is then equivalent to solving the constrained VI (1). If $F$ is well-conditioned (e.g. Lipschitz and strongly monotone) and $\mathcal{Z}$ is closed and convex, then the projected gradient method $z_{t+1} = \Pi(z_t - \eta F(z_t))$ converges to a solution $z_*$ with an appropriate stepsize $\eta$ (Facchinei & Pang, 2003).

To solve problems of the form (1) and take advantage of the structure given by $g$ and $F$, we extend the idea of surrogate losses (Johnson & Zhang, 2020; Vaswani et al., 2021), where parameters are selected by descending a sequence of surrogates $\ell_t(\theta)$ that approximate the exact projected gradient method. More precisely, at iteration $t$, $\theta_{t+1}$ is selected by descending the surrogate loss:

$$\ell_t(\theta) = \frac{1}{2} \big\| g(\theta) - [g(\theta_t) - \eta F(g(\theta_t))] \big\|^2. \tag{4}$$

Importantly, this loss and its gradient are easily constructed via automatic differentiation packages as a non-linear squared error loss. Minimizing the loss exactly gives $z_{t+1} = g(\theta_{t+1}) = \Pi(z_t - \eta F(z_t))$, the exact projected gradient step, and guarantees convergence of $\{z_t = g(\theta_t)\}_{t \in \mathbb{N}}$ to $z_*$.

**Surrogate losses in scalar minimization.** The surrogate losses proposed by Johnson & Zhang (2020) and Vaswani et al. (2021), apply to supervised learning and RL respectively.[2] They did not

---

[1] We consider the closure of the predictions to include cases such as softmax paramerizations, where the predictions only lie within the relative interior of the simplex but its closure includes the whole simplex.

[2] Note that Johnson & Zhang (2020) and Vaswani et al. (2021) more generally use Bregman divergences.

study the VI case, nor do they exploit any convexity properties in the scalar minimization case. Lavington et al. (2023) also study the scalar minimization case and provide convergence to a neighbourhood of a global minimum of (2) and allow for stochasticity. The neighbourhood of convergence depends on an upperbound on the errors $\epsilon_t = \|z_{t+1} - z_t^*\|$, where $z_t^*$ is an exact projected gradient step. Therefore, the neighbourhood of convergence scales with the worst error $\epsilon_t$ across the trajectory $\{z_t = g(\theta_t)\}_{t\in\mathbb{N}}$. Shrinking the neighbourhood necessitates a double loop algorithm that might spend too much time optimizing the surrogate. Similar results can also be found within the analysis of quasi-Fejér monotone sequences (Franci & Grammatico, 2022). For example, if $z_t^* = T(z_t)$ where $T$ is a contraction, then $\epsilon_t \to 0$ is sufficient to guarantee convergence (see Proposition A.5).

In contrast to the existing surrogate loss approaches, we propose a simple $\alpha$-descent condition on $\ell_t$ (Definition 2.1) that does not require all errors to be bounded or summable apriori. This condition allows for convergence without fully minimizing $\ell_t$ and better models implementations in practice where a fixed number of gradient descent steps are used for each $\ell_t$.

**Definition 2.1** ($\alpha$-descent). Let $\ell_t$ be the surrogate defined in (4) and $\ell_t^* = \inf_{\theta\in\mathbb{R}^d} \ell_t(\theta)$. The trajectory $\{\theta_t\}_{t\in\mathbb{N}}$ satisfies the $\alpha$-descent condition if at each step $t$ the following holds

$$\ell_t(\theta_{t+1}) - \ell_t^* \le \alpha^2\left(\ell_t(\theta_t) - \ell_t^*\right), \quad \alpha \in [0, 1). \tag{5}$$

Given the $\alpha$-descent condition we can define a general purpose algorithm (Algorithm 1). With any black-box optimizer update $\mathcal{A}$ and a double-loop structure, we can construct a trajectory that satisfies the condition so long as $\mathcal{A}$ can effectively descend the loss $\ell_t$. In general $\ell_t^*$ may not be zero and so this condition cannot be verified directly, however, this condition can often be met via first-order methods for a fixed number of steps or can be approximated with $\ell_t^* = 0$.

From a theoretical perspective, we show that the $\alpha$-descent condition guarantees the following:

$$\|z_{t+1} - z_t^*\| \le \alpha\eta\|F(z_t) - F(z_*)\|, \tag{6}$$

for a proof see Lemma A.2. As expected, the errors are controlled by $\alpha$ and the stepsize $\eta$ in the surrogate. However, the error also scales with the distance to the solution if $F$ is Lipschitz.

In the unconstrained scalar minimization case, inequality (6) is similar to the relative error condition $\|p_t - \nabla f(z_t)\| \le \alpha\|\nabla f(z_t)\|$ used in biased gradient descent, where $p_t$ is the biased direction ($z_{t+1} = z_t - p_t$) (Ajalloeian & Stich, 2020; Drusvyatskiy & Xiao, 2023). Similar to biased gradient descent, our assumption with any $\alpha < 1$ is sufficient to guarantee convergence in scalar minimization with any $L$-smooth non-convex loss $f$ (see Proposition A.3). However, for VIs, $\alpha < 1$ can still lead to divergence due to possible rotations in the dynamics of $z_t^*$ (see Proposition 3.3).

Within the context of solving VI problems, Solodov & Svaiter (1999) proposed a similar condition to approximate the proximal point algorithm where $z_t^* = z_t - \eta F(z_t^*)$.[3] However, verifying their condition would induce an inner loop with multiple evaluations of $F$, which can be expensive in many applications such as reinforcement learning (Vaswani et al., 2021). They also show that divergence is still possible under their condition unless an extra-gradient like step is performed. Unfortunately, such a step is impractical in our setting as it requires minimizing a surrogate loss exactly.

**Hidden monotone problems and preconditioning methods.** In scalar minimization problems, hidden monotone structure has been studied under hidden convexity (Bach, 2017; Xia, 2020; Chancelier & De Lara, 2021; Fatkhullin et al., 2023). In zero-sum games with hidden monotonicity, Gidel et al. (2021) study the existence of equilibria and establish approximate min-max theorems but do not propose an algorithm to take advantage of the hidden structure. For games that admit a hidden strictly convex-concave structure, Vlatakis-Gkaragkounis et al. (2021) prove global convergence of continuous time gradient descent-ascent. Similarly, Mladenovic et al. (2022) propose natural hidden gradient dynamics (NHGD) with continuous time global convergence guarantees in hidden convex-concave games. A more general and descritized version of NHGD was studied by Sakos et al. (2024), the preconditioned hidden gradient descent method (PHGD), to solve VIs of the form (1). One step of PHGD corresponds to the update:

$$\theta_{t+1} = \theta_t - \eta(Dg(\theta_t)^\top Dg(\theta_t))^\dagger Dg(\theta_t)^\top F(z_t). \tag{7}$$

---

[3]They assume $\|z_{t+1} - (z_t - \eta F(z_{t+1}))\| \le \alpha\|z_{t+1} - z_t\|$.

PHGD and stochastic variants were also studied in the linear case by Bertsekas (2009). PHGD can be viewed as a discretization of a continuous flow that guarantees $z_{t+1} = z_t - \eta F(z_t)$ as $\eta \downarrow 0$ (Sakos et al., 2024). Interestingly, in Section 4 we show that PHGD is also equivalent to taking one step of the Gauss-Newton method (GN) (Björck, 1996) on the surrogate loss.

## 3 CONVERGENCE ANALYSIS UNDER $\alpha$-DESCENT ON SURROGATE LOSSES

In this section we provide analysis in both the deterministic and stochastic settings. In both settings we make the following assumption on the hidden structure in the VI (1).

**Assumption 3.1.** In the VI (1), $\mathcal{Z}$ is convex. There exists a solution within the relative interior, $z_* \in \text{ri } \mathcal{Z}$. $F$ is both $L$-Lipschitz $\|F(x) - F(y)\| \leq L\|x - y\|$ and $\mu$-strongly monotone $\langle F(x) - F(y), x - y \rangle \geq \mu\|x - y\|^2$ for any $x, y \in \mathcal{Z}$ and some $\mu > 0$.

This assumption is commonly used in the VI literature to establish linear convergence of the projected gradient method $z_t = \Pi(z_t - \eta F(z_t))$ (Facchinei & Pang, 2003). In the scalar minimization case this assumptions implies $F(z) = \nabla f(z)$ for a scalar loss function $f$ that is $\mu$-strongly-convex and $L$-smooth. In min-max optimization with hidden structure, such as (3), Assumption (3.1) is satisfied if $f(z^1, z^2)$ is strongly-convex-concave and smooth (each players' gradient is Lipschitz with respect to all players, e.g. see Bubeck et al. (2015)). It corresponds to many practical cases as the losses used in machine learning applications are often strongly convex *with respect to the model predictions $z$*. note that $\mathcal{Z}$ is convex for many classes of models used in practice, e.g., linear models or models that can interpolate any noisy labels on the train set ($\mathcal{Z} = [0, 1]^n$). The latter has been reported with large enough neural networks (Zhang et al., 2017) and kernels (Belkin et al., 2019).

### 3.1 CONVERGENCE AND DIVERGENCE IN THE DETERMINISTIC CASE

**Convergence for sufficiently small $\alpha$.** Below we provide a linear convergence result if $\alpha$ is small enough by controlling the stepsize $\eta$ in the surrogate loss $\ell_t$. Note that the convergence is with respect to $z_t = g(\theta_t)$ and is linear in the number of outer loop iterations of Algorithm 1.

**Theorem 3.2.** *Let Assumption 3.1 hold and let $\{z_t = g(\theta_t)\}_{t \in \mathbb{N}}$ be the iterates produced by Algorithm 1. If $\alpha$ and $\eta$ are picked such that $\rho := 1 - 2\eta(\mu - \alpha L) + (1 + \alpha^2)\eta^2 L^2 < 1$ then, $z_t$ converge linearly to the solution $z_*$ at the following linear rate:*

$$\|z_{t+1} - z_*\|^2 \leq \rho^t \|z_1 - z_*\|^2. \tag{8}$$

*Particularly, if $\alpha < \frac{\mu}{L}$ and $\eta < \frac{2(\mu - \alpha L)}{(1 + \alpha^2)L^2}$ then $\rho < 1$ and if $\alpha \leq \frac{\mu}{2L}$ and $\eta = \frac{2\mu}{5L^2}$ then $\rho \leq 1 - \frac{\mu^2}{5L^2}$.*

Note that, to obtain a practical convergence rate (e.g., bounding the number of gradient computations), we need to bound the number of inner-loop steps in Algorithm 1 required to obtain the condition (5). In general, we cannot provide any global guarantee as it is well established that, in general, finding a global minima of a smooth non-convex optimization function like (4) can be intractable (Murty & Kabadi, 1985; Nemirovskij & Yudin, 1983). However, many classes of over-parametrized models are known to be able to *interpolate* random labels (e.g., neural networks (Zhang et al., 2017), kernels (Belkin et al., 2019), or boosting (Bartlett et al., 1998)) which is strong evidence that, in practice, the condition (5) can be obtained with a few gradient steps. The latter is also supported by our experiments in Section 5.

**Divergence with $\alpha < 1$.** In *non-convex* smooth scalar minimization, $\alpha < 1$ is sufficient for convergence (Proposition A.3). However, despite our strong monotonicity assumption we show that $\alpha < 1$ can still give divergence in the VI setting. This shows that $\alpha$ being small enough is not only sufficient but is *necessary* for convergence. Our example demonstrates the additional challenges of using a surrogate loss in the VI setting. Our construction uses the min-max problem $\min_x \max_y \frac{1}{2}x^2 + xy - \frac{1}{2}y^2$ and showing that gradient descent-ascent on the loss $f(x, y) = xy$ satisfies the $\alpha$-descent condition with $\alpha = 1/\sqrt{2}$ but diverges for any $\eta$.

**Proposition 3.3.** *There exists an $L$-Lipschitz and $\mu$-strongly monotone $F$, and a sequence of iterates $\{z_t\}_{t \in \mathbb{N}}$ verifying the alpha descent condition with $\alpha < 1$ such that $z_t$ diverges for any $\eta$.*

## 3.2 Unconstrained Stochastic Case

For the stochastic case we assume $\mathcal{Z} = \mathbb{R}^n$, the predictions can represent all of $\mathbb{R}^n$. Although this assumption is strong it can be satisfied for large capacity neural networks that can interpolate any dataset. The stochastic case is more challenging as our setup corresponds to solving VIs with bias, for which we show can diverge even in the deterministic case (Proposition 3.3).

Since $\mathcal{Z} = \mathbb{R}^n$, no projection is needed for $z_t^*$, and the minimum of the surrogate $\ell_t$ is the gradient step $z_t^* = z_t - \eta F(z_t)$. We also assume the standard setup, the possibility to generate independent and identically distributed realizations $(\xi_1, \xi_2, \cdots)$ of a random variable $\xi$ such that $F_\xi(z)$ is an unbiased estimator of $F(z)$. For the noise, we define $\sigma^2 = \mathbb{E}_\xi \left[ \|F_\xi(z_*)\|^2 \right]$ and use the expected co-coercive assumption from Loizou et al. (2021):

**Assumption 3.4** (expected co-coercivity[4]). $\mathbb{E}_\xi \left[ \|F_\xi(z)\|^2 \right] \leq 2L\langle F(z), z - z_* \rangle + 2\sigma^2$, for all $z \in \mathcal{Z}$.

Without access to the true operator $F$ in the stochastic case, a different $\alpha$-descent condition is required using the stochastic estimate $F_{\xi_t}(z_t)$. Instead of $\ell_t$ (4), we will use the loss $\tilde{\ell}_t$, an approximation of a projected stochastic gradient step, $\tilde{\ell}_t(\theta) = \frac{1}{2}\|g(\theta) - (g(\theta_t) - \eta F_{\xi_t}(z_t))\|^2$, or equivalently

$$\tilde{\ell}_t(\theta) = \frac{\eta^2}{2}\|F_{\xi_t}(z_t)\|^2 + \eta\langle F_{\xi_t}(z_t), g(\theta) - g(\theta_t)\rangle + \frac{1}{2}\|g(\theta) - g(\theta_t)\|^2. \tag{9}$$

In practice, minimizing $\tilde{\ell}_t$ exactly is impractical if $n$ is very large due to the sum of squares $\|g(\theta) - g(\theta_t)\|^2$ in (9). Therefore we assume it can be minimized on expectation with the following assumption.

**Definition 3.5** ($\alpha$-expected descent). The trajectory $\{\theta_t\}_{t\in\mathbb{N}}$ satisfies the $\alpha$-expected descent condition if at for each $t$ the following holds: given $F_{\xi_t}(z_t)$, the parameter $\theta_{t+1}$ is generated such that

$$\mathbb{E}\left[\tilde{\ell}_t(\theta_{t+1}) - \tilde{\ell}_t^*\right] \leq \alpha^2(\tilde{\ell}_t(\theta_t) - \tilde{\ell}_t^*).$$

Similar to the deterministic case, we can prove linear convergence to a neighbourhood.

**Theorem 3.6.** *Let $\mathcal{Z} = \mathbb{R}^n$, and Assumption (3.4) hold. If $F_\xi(x)$ is $L$-Lipschitz and $\eta \leq \frac{1}{2(1+c)L}$ where $c \geq 2(1 + \alpha^2)$ then any trajectory $\{\theta_t\}_{t\in\mathbb{N}}$ satisfying the $\alpha$-expected descent condition guarantees:*

$$\mathbb{E}\left[\tfrac{1}{2}\|z_{t+1} - z_*\|^2\right] \leq \mathbb{E}\left[\tfrac{1}{2}\|z_t - z_*\|^2\right]\left(1 - \eta\mu + \alpha^2\right) + \eta^2(1 + c)\sigma^2.$$

If we take $c = 4$ and $\eta \leq 1/10L$ then we have convergence to a neighbourhood if $\alpha < \sqrt{\eta\mu}$. Theorem 3.6 provides a generalization of Loizou et al. (2021)[Theorem 4.1] where $z_{t+1}$ follows a random biased direction $p_t$ that on expectation guarantees $\mathbb{E}\left[\|p_t - \eta F_{\xi_t}(z_t)\|^2\right] \leq \alpha^2\eta^2\|F_{\xi_t}(z_t)\|^2$. In comparison to the stochastic results of Lavington et al. (2023), they avoid the assumption $\mathcal{Z} = \mathbb{R}^n$ by using a different surrogate. However, their result only applies to scalar minimization.

## 4 A Nonlinear Least Squares Perspective

The surrogate loss perspective and our $\alpha$-descent condition allows for convergence so long as the surrogate losses $\{\ell_t\}_{t\in\mathbb{N}}$ are sufficiently minimized. One approach to minimizing $\ell_t$ is to view it as the following non-linear least-squares problem

$$\min_\theta f(\theta) = \min_\theta \frac{1}{2}\|r(\theta)\|^2, \tag{10}$$

with a residual function $r : \mathbb{R}^d \to \mathbb{R}^n$, where $\ell_t(\theta) = f(\theta)$ if $r(\theta) = g(\theta) - g(\theta_t) + \eta F(g(\theta_t))$. Due to the specific structure of $f$ we can consider specialized methods such as Gauss-Newton (GN), Damped Gauss-Newton (DGN), and Levenbergh-Marquardt (LM) (Björck, 1996; Nocedal & Wright, 1999). These methods can be viewed as quasi-Newton methods that use a linear approximation of $r$, $r(\theta) \approx r(\theta_t) + Dr(\theta_t)(\theta - \theta_t)$.

The GN method is defined by the update rule $\theta_{t+1} = \theta_t - (Dr(\theta_t)^\top Dr(\theta_t))^\dagger Dr(\theta_t)^\top r(\theta_t)$. GN inherits the same local quadratic convergence properties as Newton's method when the Hessian at

---

[4]Note that this is not exactly expected co-coercivity but implied by it (Loizou et al., 2021, Lemma 3.4).

the minimum $\nabla^2 f(\theta_*) \approx Dr(\theta_*)^\top Dr(\theta_*)$. However, GN is known to struggle with highly non-linear problems, those with large residuals, or if $Dr(\theta_t)$ is nearly rank-deficient (Björck, 1996). Fortunately, the GN direction is a descent direction of $f$, the DGN method takes steps in the GN direction with a stepsize parameter $\eta_{GN}$ and converges for a sufficiently small stepsize or with line search (Björck, 1996). In cases where $Dr(\theta_t)$ is nearly rank deficient, the LM method can be used.

To minimize the surrogate we can consider taking multiple steps of gradient descent (Surr-GD), DGN or LM. Denoting $\theta_t^s$ as $s^{\text{th}}$ intermediate step between $\theta_{t+1}$ and $\theta_t$ we have the following:

$$\theta_t^{s+1} = \theta_t^s - \eta_{GD} Dg(\theta_t^s)^\top (g(\theta_t^s) - g(\theta_t) + \eta F(g(\theta_t))) \tag{Surr-GD}$$

$$\theta_t^{s+1} = \theta_t^s - \eta_{GN}(Dg(\theta_t^s)^\top Dg(\theta_t^s))^\dagger Dg(\theta_t^s)^\top (g(\theta_t^s) - g(\theta_t) + \eta F(g(\theta_t))) \tag{DGN}$$

$$\theta_t^{s+1} = \theta_t^s - (Dg(\theta_t^s)^\top Dg(\theta_t^s) + \lambda \operatorname{Id})^{-1} Dg(\theta_t^s)^\top (g(\theta_t^s) - g(\theta_t) + \eta F(g(\theta_t))). \tag{LM}$$

Note that we used the fact that $Dr(\theta) = Dg(\theta)$, and if $\eta_{GN} = 1$ then DGN is the same as GN. Also, note that one step of GN recovers exactly the PHGD method proposed by Sakos et al. (2024).

## 4.1 FAVOURABLE CONDITIONS FOR GRADIENT DESCENT

Several conditions allow for fast linear convergence of gradient descent for $\ell_t(\theta) - \ell_t^*$; see for example Guille-Escuret et al. (2021). Under linear convergence, $s$-steps of gradient descent guarantees $\alpha = \rho^s$ for some $\rho \in [0, 1)$, therefore any target value of $\alpha$ is achievable in a finite number of steps that depends only logarithmically in $\alpha$. In this section, we show how assumptions used in Sakos et al. (2024) imply two common assumptions for fast convergence of GD: the Polyak-Łojasiewicz (PL) condition (Polyak, 1964; Łojasiewicz, 1963, Definition B.2) and $L$-smoothness.

The four assumptions made by Sakos et al. (2024) to study the behaviour of PHGD are: (1-2) $Dg^\top$ has both uniformly lower bounded and upper bounded singular values, (3) each component function $g_i$ is $\beta$-smooth, (4) and that $\mathcal{Z}$ is bounded.[5] In Proposition 4.1 we show that globally lower bounding the singular values of $Dg^\top$ guarantees that the composition of a PL function $f$ with $g$ is still PL.

**Proposition 4.1.** *Assume $f$ satisfies the $\mu$-PL condition, and let $\sigma_{\min}$ be a lower bound on the singular values of $Dg(\theta)^\top$. Then, $f \circ g$ is $\mu \sigma_{\min}^2$-PL.*

This implies that the surrogate loss $\ell_t$ is PL since $\ell_t = f_t \circ g$ where $f_t(z) = \frac{1}{2}\|z - v_t\|^2$ is 1-PL. Similarly, $\ell_t$ can be shown to be $L$-smooth if $Dg^\top$ has uniformly upper bounded singular values, each $g_i$ is $\beta$-smooth, and $\mathcal{Z}$ is bounded (see Lemma B.1). However, these 4 assumptions cannot hold all at once. In Prop. 4.2, we show that if the first three assumptions (1-3) hold, then $\mathcal{Z}$ is unbounded.

**Proposition 4.2.** *If $g : \mathbb{R}^d \to \mathbb{R}^n$ is differentiable where $g_i$ is $\beta$-smooth and $Dg(\theta)^\top$ has globally lower and upper bounded singular values, then $\{g(\theta) : \theta \in \mathbb{R}^d\}$ is unbounded.*

This contradiction suggests that the lower bounded singular value assumption is strong. Indeed, it forces the dimension of the parameter space $d$ to be larger than the dimension of the prediction space $n$, and if $g$ is linear then it must be surjective $\mathcal{Z} = \mathbb{R}^n$. In the case $n = d$, it enforces $Dg$ to be invertible everywhere, which is violated in important cases such as softmax.

## 5 EXPERIMENTS

We consider min-max optimization and policy evaluation to demonstrate the behaviour and convergence of surrogate based methods with hidden monotone structure. For min-max optimization we compare different approaches from Section 4 on two domains from Sakos et al. (2024): hidden matching pennies and hidden rock-paper scissors. For RL policy evaluation we consider the problem of minimizing projected Bellman error (PBE) with linear and non-linear approximation.

## 5.1 MIN-MAX EXPERIMENTS

We compare four different approaches from Section 4: GN, DGN, LM, and Surr-GD. Taking only one inner step for GN and Surr-GD gives PHGD and gradient descent-ascent (GDA), respectively.

---

[5]The set $\mathcal{Z}$ (note that $\mathcal{Z}$ is the "set of latent variables" denoted as $\mathcal{X}$ in Sakos et al. (2024)) is assumed to be bounded in their Lemma 4 "template inequality". More precisely, the existence of a constant $D = diam(\mathcal{X})$ (i.e. $\mathcal{X}$ is bounded) is used in equation B.20 in Appendix B. Lemma 4 is then used in Theorems 1-4.

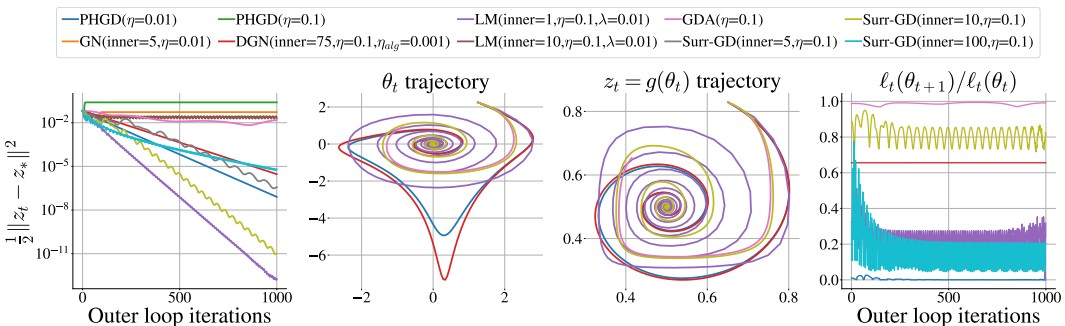

Figure 1: Convergence of various algorithms from Section 4 on the hidden matching pennies game. PHGD and GDA as presented in Sakos et al. (2024) are compared against GN, DGN, LM, and GD. (left) Linear convergence to the equilbrium is observed for several methods with LM and GD outperforming the rest. (middle) Trajectories for some methods are plotted in both the parameter and prediction space. (right) The loss ratio $\ell_t(\theta_{t+1})/\ell_t(\theta_t)$ is illustrated for the considered methods.

**Hidden Matching Pennies.** The hidden matching pennies game is a zero-sum game of the form (3), where each player has the parameterization $h^i(\theta) = \mathrm{sigmoid}(\alpha_2^i \mathrm{CELU}(\alpha_1^i \theta))$. The convex-concave objective $f$ is given by $f(z^1, z^2) = -(2z^1 - 1)(2z^2 - 1) + \frac{0.75}{2}\left((z^1 - 1/2)^2 + (z^2 - 1/2)^2\right)$. The parameters $\alpha_j^i$ are chosen to approximately replicate the trajectory of PHGD presented in Sakos et al. (2024, Figure 4) (see Appendix C.1 for more details). In Figure 1, we observe that PHGD converges linearly

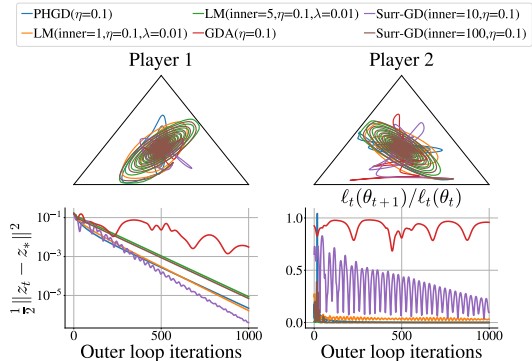

Figure 2: Convergence in the hidden rps game.

in the squared distance to the equilibrium, but performs poorly if multiple inner steps are taken (GN with 5 inner steps). If $\eta$ is increased by an order of magnitude, PHGD is observed to diverge; however, convergence is possible via DGN with the same $\eta$, with multiple iterations and an appropriate choice of $\eta_{DGN}$. In contrast to GN, LM is more stable with a larger $\eta$ and converges faster than PHGD/GN. Finally we tested GD for a different number of inner steps with $\eta = 0.1$. Convergence is observed for GDA albeit slow. The benefit of multiple steps is clear, with 10 inner steps outperforming PHGD and only surpassed by LM. Although more inner steps increases the computational cost, it is marginal when compared to evaluating $F$ (see Fig. 5). Interestingly, Fig. 1 (right) shows that spending more compute to minimize the surrogate at each iteration does not necessarily translate to faster overall convergence with respect to the outer loop (left). GD with 10 inner steps has a larger loss ratio than GD with 100 steps but converges faster to the equilibrium.

**Hidden rock-paper-scissors.** In the hidden rps game each player's mixed strategy in rock-paper-scissors is parameterized. Where player $i$'s strategy $z^i$ is given by the function $h^i(\theta) = \mathrm{softmax}(A_2^i \mathrm{CELU}(A_1^i \theta^i))$, with $\theta^i \in \mathbb{R}^5$ and randomly initialized matrices: $A_1^i \in \mathbb{R}^{4 \times 5}$, and $A_2^i \in \mathbb{R}^{3 \times 4}$. Figure 2 demonstrates the behaviour of various algorithms for a fixed initialization of $\theta = (\theta^1, \theta^2)$ and the matrices $A_j^i$. We observe that PHGD and LM with one inner step achieve linear convergence while GDA performs poorly with an unstable behaviour. Like in the hidden matching pennies game, increasing the number of inner steps for GD improves stability and performance, with the best performance not necessarily corresponding to the methods with the lowest loss ratio. Both LM and GD degrade in performance if too many inner steps are taken, with one and 10 inner steps outperforming 5 and 100 steps for LM and GD respectively.

## 5.2 MINIMIZING PROJECTED BELLMAN ERROR

For our RL experiments we consider the policy evaluation problem of approximating a value vector $v_\pi \in \mathbb{R}^n$, representing the expected discounted return over $n$ states for a given policy $\pi$ and Markov decision process (MDP). To this end, we consider a common approach to policy evaluation: minimizing the projected Bellman error, which is a fixed point problem associated with temporal difference learning (TD) (Sutton, 1988). Despite the widespread practical success of TD and its variants, the analysis and behaviour of TD has proven challenging since it cannot be modeled as a scalar minimizing problem and is known not to follow the gradient of any objective (Baird, 1995;

Antos et al., 2008). Bertsekas (2009) showed that minimizing PBE is equivalent to solving a smooth and strongly monotone VI problem of the form (1) where $F(z) = \Xi(z - T_\pi(z))$ (Bertsekas, 2009). $T_\pi : \mathbb{R}^n \to \mathbb{R}^n$ is the linear Bellman policy evaluation operator defined over all $n$ states $(s^1, \cdots, s^n)$ in a given MDP and is defined as $T_\pi(z) = r_\pi + \gamma P_\pi z$, where $r_\pi$ is the associated expected reward vector at each state and $P_\pi$ is the state to state probability transition matrix as given by the MDP and $\pi$. $\Xi$ is the diagonal matrix with entries corresponding to the stationary distribution $\xi \in \mathbb{R}^n$ of states according to $\pi$. The constraint $\mathcal{Z}$ in this case is the set of all representable value functions. With respect to $\theta$, solving the VI associated with minimizing PBE is equivalent to finding the fixed point:

$$\theta_* \in \arg\min_\theta \left[ BE(\theta, \theta_*) = \frac{1}{2} \sum_{i=1}^n \xi^i (v_\theta(s^i) - (r_\pi(s^i) - \gamma \mathbb{E}_{s' \sim P_\pi} \left[ v_{\theta_*}(s')|s^i \right])^2 \right], \qquad (11)$$

where $v_\theta$ is the predicted value vector given parameters $\theta$. See Appendix D for more details.

### 5.2.1 Linear Approximation

For our linear RL experiments, we consider a slow-mixing 100-state Markov chain from Bertsekas (2009) and Yu & Bertsekas (2009). Using a linear model, $z = g(\theta) = \Phi\theta$ with $\theta \in \mathbb{R}^d$, $\Phi \in \mathbb{R}^{n \times d}$ and $d \ll n$. Bertsekas (2009) proposed several methods to leverage the hidden structure in $F$. Similar to Sakos et al. (2024), the proposed methods are presented via preconditioning schemes that approximate a projected gradient step in the space of representable value functions $\mathcal{Z}$. Bertsekas (2009) suggests performing the deterministic and stochastic PHGD-like updates:

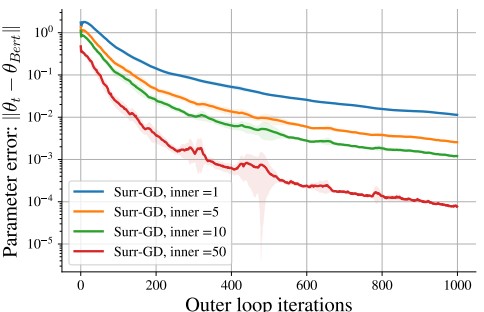

Figure 3: The average approximation error between GD on the surrogate (14, Surr-GD) and update (13) over 10,000 runs in a slow mixing 100-state Markov chain from Bertsekas (2009) and Yu & Bertsekas (2009, Example 3). Surr-GD is observed to converge to the exact update (13) with faster convergence for more inner steps.

$$\theta_{t+1} = \theta_t - (\Phi^\top \Xi \Phi)^{-1} \Phi^\top F(z_t), \qquad (12)$$

$$\theta_{t+1} = \theta_t - \hat{D}_t^{-1}(\hat{C}_t \theta_t - \hat{r}_t). \qquad (13)$$

The deterministic update is expensive, requiring the full feature matrix $\Phi$. Hence, Bertsekas (2009) constructs an online sequence of estimators $\{\hat{D}_t\}_{t \in \mathbb{N}}$, $\{\hat{C}_t\}_{t \in \mathbb{N}}$, and $\{\hat{r}_t\}$ from a trajectory and suggests the stochastic update (13). Although this method applies more generally, in the context of solving (11) update (13) is equivalent to least-squares policy evaluation (Bertsekas & Ioffe, 1996).

Equivalently, update (12) can be seen as one step of the generalized Gauss-Newton method (Ortega & Rheinboldt, 2000) on the surrogate $\ell_t(\theta) = 1/2\|\Phi\theta - T_\pi(\Phi\theta_t)\|_\Xi^2$. Since the model is linear, the update is the minimizer of the surrogate, and corresponds to an exact projected gradient step in the space of value functions, which is known to be a contraction (Bertsekas, 2012).

Similar to the deterministic update, we show that the stochastic version can be shown to be the minimizer of a surrogate loss $\tilde{\ell}_t(\theta)$, whose minimizer is the exact update (13). The stochastic surrogate $\tilde{\ell}_t$ and its gradient, derived in Proposition D.1, are

$$\tilde{\ell}_t(\theta) = \frac{1}{2}\|\Phi\theta - \hat{T}_\pi(z_t)\|_{\hat{\Xi}}^2, \qquad \nabla \tilde{\ell}_t(\theta) = \hat{C}_t \theta_t - \hat{r}_t + \hat{D}_t(\theta - \theta_t), \qquad (14)$$

where $\hat{\Xi}$ and $\hat{T}_\pi$ correspond to the natural estimators using the empirical distribution (see Appendix D.1 for more details). Instead of minimizing the loss exactly it can be approximated with GD, thus avoiding the matrix inversion $\hat{D}_t^{-1}$, which can make (13) intractable. Figure 3 demonstrates the effectiveness of approximating update (13) via GD on the surrogate $\tilde{\ell}_t$ (14).

### 5.2.2 Non-linear Projected Bellman Error

For the non-linear setting, we approximate the value function of a fixed reference policy with a two-layer neural network in two Mujoco environments, Ant and Half Cheetah (Todorov et al., 2012). These environments are characterized by continuous state and action spaces, where the agent must

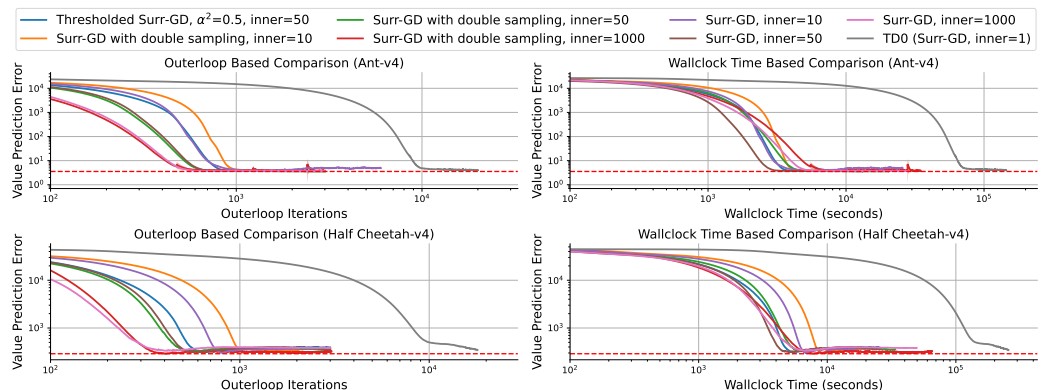

Figure 4: Comparison of average performance of TD(0) and surrogate methods in minimizing the value prediction error for RL tasks with nonlinear function approximation in Ant (top) and HalfCheetah (bottom) environments, measured by outer loop iterations (left) and wallclock time (right). The average value prediction error across 20 runs along with 95% confidence intervals are computed from a fixed test set. The red dashed line represents the lowest value prediction error achieved by any of the algorithms.

control high-dimensional, physics-based simulations to achieve locomotion tasks. The reference policy is derived by training a policy network with the soft-actor-critic algorithm (Haarnoja et al., 2018). For each environment, we consider two different surrogate-based methods with TD(0) as a special case. Similar to the linear case, we aim to find the fixed point (11), and consider the surrogate $\ell_t(\theta) = 1/2\|v_\theta - T_\pi(v_{\theta_t})\|_\Xi^2$. However, due to the large state spaces we must instead consider minimizing the stochastic surrogate $\tilde{\ell}_t$ (9) from Section 3.2. Notice that $\tilde{\ell}_t$ consists of two main parts: an inner product term that is linear in $v_\theta = g(\theta)$, and a squared error term $\|v_\theta - v_{\theta_t}\|^2$. Our two surrogate methods use different estimates for the linear and error term.

The first method, Surr-GD, uses the same batch of data to approximate both the inner product part and the squared error term. This fixed batch approach was proposed in Lavington et al. (2023) for scalar minimization, and introduces bias due to approximating the squared error term across all states with states seen in the batch. With this approximation we perform either a fixed number of steps (inner = # steps) or an adaptive number of steps depending on a loss ratio (Thresholded Surr-GD). Taking one step recovers a batch version of TD(0), for more details see Algorithms 2, 4.

The second method, surrogate with double sampling, uses a fixed batch to approximate the inner product term but resamples a new batch to approximate the squared error part at each gradient step. By resampling a new batch for the squared error we have removed the bias from the first method at the cost of increased variance in the gradient. For more details see Algorithm 3.

All methods are evaluated with value prediction error, the squared error for a test sample of 10,000 states visited by a fixed reference policy. The values in these test states are approximated by 10 Monte-Carlo roll-outs. As shown in Figure 4, surrogate methods significantly outperformed TD(0) in both data efficiency (outer loop iterations) and wall-clock time. Interestingly, the surrogate methods converge to a lower prediction error than TD(0), as indicated by the red dashed line.

Although the inner loop in the surrogate methods requires more gradient computations, the computational overhead is marginal when compared to environment interactions, a common bottleneck in RL tasks (Vaswani et al., 2021). We observe that increasing the inner loop size improves data efficiency at the trade-off of a potentially slower convergence in wall-clock time. An inner loop count, such as 50, seems to balance both data efficiency and wall-clock time.

## 6 CONCLUSION

In this work we proposed a principled and scalable surrogate loss approach to solving variational inequality problems with hidden monotone structure. We have presented a novel $\alpha$-descent condition that is optimizer agnostic and quantifies sufficient progress on the surrogate for convergence. We have proved linear convergence in both the deterministic and stochastic settings. We have also shown that surrogate losses in VI problems are strictly more difficult to analyze than in scalar minimization. Furthermore, we have demonstrated the generality of our approach by showing how existing methods can be viewed as special cases of our general framework. Empirically, we have demonstrated the effectiveness of using surrogate losses in both min-max optimization and minimizing projected Bellman error. Finally, by using surrogate losses, we have proposed novel variants of TD(0) that are more data and run-time efficient in the deep reinforcement learning setting.

ACKNOWLEDGMENTS

We thank Nicolas Le Roux, David Kanaa, Iosif Sakos, Andrew Patterson, Khurram Javed, and Arushi Jain, for helpful discussions and feedback. This work was supported by Borealis AI through the Borealis AI Global Fellowship Award.

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

# A PROOFS

**Remark A.1.** If $g$ is continuous, and $\{g(\theta) : \theta \in \mathbb{R}^d\}$ is convex with $\mathcal{Z}$ as its closure, then the least-squares surrogate loss $\ell_t(\theta) = \frac{1}{2}\|g(\theta) - z_t + \eta F(z_t)\|^2$ admits a unique point $z_t^* \in \mathcal{Z}$ such that

$$\ell_t^* = \frac{1}{2}\|z_t^* - z_t + \eta F(z_t)\|^2,$$

and for any $\theta$

$$\frac{1}{2}\|g(\theta) - z_t^*\|^2 \le \ell_t(\theta) - \ell_t^*,$$

where $\ell_t^* = \inf_{\theta \in \mathbb{R}^d} \ell_t(\theta)$.

*Proof.* Let $f(z) = \frac{1}{2}\|z - z_t + \eta F(z_t)\|^2$ be the surrogate loss with respect to the predictions $z = g(\theta)$. We have that $f(z) = \ell_t(\theta)$ for all $\theta \in \mathbb{R}^d$. Now consider the set $\mathcal{Z} = \mathrm{cl}\{g(\theta) : \theta \in \mathbb{R}^d\}$, since it is closed and convex, we have that $f$ has a unique minimum $z_t^*$ because it is 1-strongly convex. Furthermore, we have

$$\frac{1}{2}\|z - z_t^*\|^2 \le f(z) - f(z_t^*).$$

Now since $z_t^* \in \mathcal{Z}$ and $\mathcal{Z}$ is the closure of $\{g(\theta) : \theta \in \mathbb{R}^d\}$, there exists a sequence of parameters $\{\theta_t\}_{t \in \mathbb{N}}$ such that $\{z_t = g(\theta_t)\}_{t \in \mathbb{N}} \to z_t^*$. Therefore we have that,

$$f(z_t^*) = \lim_{t \to \infty} f(z_t) = \lim_{t \to \infty} \ell_t(\theta_t) \ge \ell_t^* \ge f(z_t^*).$$

Where we have used the continuity of $f$ and $g$. The last inequality follows because $\{g(\theta) : \theta \in \mathbb{R}^d\} \subseteq \mathcal{Z}$. Therefore $\ell_t^* = f(z_t^*)$ and the result follows. $\square$

**Lemma A.2.** *If $F$ is monotone and $z_*$ is in the relative interior of the constraint $\mathcal{Z}$, then*

$$\ell_t(\theta_t) - \ell_t^* \le \frac{\eta^2}{2}\|F(z_t) - F(z_*)\|^2.$$

*Furthermore, under the $\alpha$-descent condition (Definition 2.1) we have*

$$\|z_{t+1} - z_t^*\| \le \alpha\eta\|F(z_t) - F(z_*)\|.$$

*Proof.* If $z_*$ is a solution then we have

$$\langle F(z_*), z - z_* \rangle \ge 0\,, \ \forall z \in \mathcal{Z}.$$

If $z_*$ is in the relative interior then for any $z \in \mathcal{Z}$ there exists a $\lambda > 1$ such that $z' = (1-\lambda)z + \lambda z_* \in \mathcal{Z}$ (Rockafellar, 1997)[Theorem 6.4]. Therefore by optimality we have

$$0 \le \langle F(z_*), z' - z_* \rangle = \langle F(z_*), (1-\lambda)z + \lambda z_* - z_* \rangle = (1 - \lambda)\langle F(z_*), z - z_* \rangle \le 0.$$

Where the last inequality follows from $\lambda > 1$. Altogether we have

$$\langle F(z_*), z - z_* \rangle = 0\,, \ \forall z \in \mathcal{Z}.$$

Moreover, as a consequence we have that for any two points $z, z' \in \mathcal{Z}$

$$\langle F(z_*), z - z' \rangle = \langle F(z_*), z - z_* \rangle + \langle F(z_*), z_* - z' \rangle = 0.$$

Letting $z_t^*$ be the exact projected gradient step and $z_t = g(\theta_t)$ the current iterate we have

$$\ell_t(\theta_t) - \ell_t^* = \frac{1}{2}\|\eta F(z_t)\|^2 - \left(\frac{1}{2}\|z_t^* - z_t + \eta F(z_t)\|^2\right)$$

$$= \eta\langle F(z_t), z_t - z_t^* \rangle - \frac{1}{2}\|z_t^* - z_t\|^2$$

$$= \eta\langle F(z_t) - F(z_*), z_t - z_t^* \rangle + \eta\langle F(z_*), z_t - z_t^* \rangle - \frac{1}{2}\|z_t^* - z_t\|^2$$

$$= \eta\langle F(z_t) - F(z_*), z_t - z_t^* \rangle - \frac{1}{2}\|z_t^* - z_t\|^2$$

$$\le \frac{\eta^2}{2}\|F(z_t) - F(z_*)\|^2.$$

Where the last two inequalities follow by: $z_*$ being within the relative interior, and the inequality $\langle u, v \rangle \leq \frac{\rho}{2}\|u\|^2 + \frac{1}{2\rho}\|v\|^2$ for any $\rho > 0$.

By Remark A.1 and the $\alpha$-decent condition,

$$\frac{1}{2}\|z_{t+1} - z_t^*\|^2 \leq \ell_t(\theta_{t+1}) - \ell_t^* \leq \alpha^2(\ell_t(\theta_t) - \ell_t^*) \leq \alpha^2\frac{\eta^2}{2}\|F(z_t) - F(z_*)\|^2.$$

$\square$

**Proposition A.3.** *Let* $f : \mathbb{R}^n \to \mathbb{R}$ *be L-smooth. For some* $g : \mathbb{R}^d \to \mathbb{R}^n$ *define the surrogate loss* $\ell_t(\theta) = \frac{1}{2}\|g(\theta) - z_t + \frac{1}{L}\nabla f(z_t)\|^2$, *where* $z_t = g(\theta_t)$. *Then the* $\alpha$-*descent condition (Definition 2.1) guarantees*

$$f(z_{t+1}) \leq f(z_t) - L(1 - \alpha^2)(\ell_t(\theta_t) - \ell_t^*).$$

*If* $f$ *is bounded below then* $\{z_t - z_t^*\}_{t\in\mathbb{N}} \to 0$, *where* $z_t^* = \Pi(z_t - \frac{1}{L}\nabla f(z_t))$.

*Proof.* Let $\hat{z}_t = z_t - \frac{1}{L}\nabla f(z_t)$.

$$\begin{aligned}
f(z_{t+1}) - f(z_t) &\leq \langle \nabla f(z_t), z_{t+1} - z_t \rangle + \frac{L}{2}\|z_{t+1} - z_t\|^2 \\
&= L\langle z_t - \hat{z}_t, z_{t+1} - z_t \rangle + \frac{L}{2}\|z_{t+1} - z_t\|^2 \\
&= L\left(-\|\hat{z}_t - z_t\|^2 + \langle z_t - \hat{z}_t, z_{t+1} - \hat{z}_t \rangle + \frac{1}{2}\|z_{t+1} - z_t\|^2\right) \\
&= L\left(\frac{1}{2}\|\hat{z}_t - z_{t+1}\|^2 - \frac{1}{2}\|\hat{z}_t - z_t\|^2\right) \\
&= L\left(\ell_t(\theta_{t+1}) - \ell_t(\theta_t)\right).
\end{aligned}$$

The second to last equality follows from expanding $\frac{1}{2}\|z_{t+1} - z_t\|^2 = \frac{1}{2}\|z_{t+1} - \hat{z}_t + \hat{z}_t - z_t\|^2$. Using the $\alpha$-descent condition we have

$$\ell_t(\theta_{t+1}) - \ell_t(\theta_t) \leq (\alpha^2 - 1)(\ell_t(\theta_t) - \ell_t^*),$$

yielding the first result.

If $f$ is bounded below by some constant $c$ then we have:

$$\sum_{t=1}^{T} L(1 - \alpha^2)(\ell_t(\theta_t) - \ell_t^*) \leq f(z_1) - f(z_T) \leq f(z_1) - c.$$

Therefore the series $\sum_{t=1}^{\infty} \ell_t(\theta_t) - \ell_t^*$ converges with $\ell_t(\theta_t) - \ell_t^* \to 0$. By Remark A.1 we have $\frac{1}{2}\|z_t - z_t^*\|^2 \leq \ell_t(\theta_t) - \ell_t^*$ implying the result $z_t - z_t^* \to 0$. $\square$

**Theorem 3.2.** *Let Assumption 3.1 hold and let* $\{z_t = g(\theta_t)\}_{t\in\mathbb{N}}$ *be the iterates produced by Algorithm 1. If* $\alpha$ *and* $\eta$ *are picked such that* $\rho := 1 - 2\eta(\mu - \alpha L) + (1 + \alpha^2)\eta^2 L^2 < 1$ *then,* $z_t$ *converge linearly to the solution* $z_*$ *at the following linear rate:*

$$\|z_{t+1} - z_*\|^2 \leq \rho^t\|z_1 - z_*\|^2. \tag{15}$$

*Particularly, if* $\alpha < \frac{\mu}{L}$ *and* $\eta < \frac{2(\mu - \alpha L)}{(1 + \alpha^2)L^2}$ *then* $\rho < 1$ *and if* $\alpha \leq \frac{\mu}{2L}$ *and* $\eta = \frac{2\mu}{5L^2}$ *then* $\rho \leq 1 - \frac{\mu^2}{5L^2}$.

*Proof.* First note that by definition of Algorithm 1, the iterates $z_t = g(\theta_t)_{t\in\mathbb{N}}$ satisfy the $\alpha$-descent property (Definition 2.1), therefore Lemma A.2 holds. Recall that $z_t^*$, the exact projection update, is a contraction if $\eta < \frac{2\mu}{L}$ since

$$\|z_t^* - z_*\|^2 \leq \kappa^2\|z_t - z_*\|^2 \tag{16}$$

where $\kappa^2 = 1 - 2\eta\mu + \eta^2 L^2$(Facchinei & Pang, 2003, Theorem 12.1.2). For the remainder, assume that $\eta < \frac{2\mu}{L}$ so that $\kappa \in [0, 1)$. Denoting $F_t = F(z_t)$ and $F_* = F(z_*)$, we have

$$
\begin{aligned}
\|z_{t+1} - z_*\|^2 &= \|z_t^* - z_* + z_{t+1} - z_t^*\|^2 \\
&= \|z_t^* - z_*\|^2 + 2\langle z_{t+1} - z_t^*, z_t^* - z_* \rangle + \|z_{t+1} - z_t^*\|^2 \\
&\leq \|z_t^* - z_*\|^2 + 2\|z_{t+1} - z_t^*\|\|z_t^* - z_*\| + \|z_{t+1} - z_t^*\|^2 \quad \text{(Cauchy–Schwarz)} \\
&\leq \|z_t^* - z_*\|^2 + 2\alpha\eta\kappa\|F_t - F_*\|\|z_t - z_*\| + \alpha^2\eta^2\|F_t - F_*\|^2 \quad \text{(Lemma A.2)} \\
&\leq \|z_t^* - z_*\|^2 + 2\alpha\eta L\|z_t - z_*\|^2 + \alpha^2\eta^2 L^2\|z_t - z_*\|^2 \quad \text{(Smoothness of $F$ and $\kappa < 1$)} \\
&\leq \kappa^2\|z_t - z_*\|^2 + 2\alpha\eta L\|z_t - z_*\|^2 + \alpha^2\eta^2 L^2\|z_t - z_*\|^2 \quad \text{(Eq. 16)} \\
&= \|z_t - z_*\|^2 \left( 1 - 2\eta\mu + 2\alpha\eta L + (1 + \alpha^2)\eta^2 L^2 \right).
\end{aligned}
$$

If $\alpha < \frac{\mu}{L}$ then

$$
\|z_{t+1} - z_*\|^2 \leq \|z_t - z_*\|^2 \left( 1 - 2\eta \underbrace{(\mu - \alpha L)}_{>0} + (1 + \alpha^2)\eta^2 L^2 \right).
$$

Taking $\eta < \frac{2(\mu - \alpha L)}{(1 + \alpha^2)L^2}$ would guarantee a contraction.

If $\alpha \leq \frac{\mu}{2L}$ and taking $\eta = \frac{2\mu}{5L^2}$ we have:

$$
1 - 2\eta\mu + 2\alpha\eta L + (1 + \alpha^2)\eta^2 L^2 \overset{\alpha \leq \mu/2L \leq 1/2}{\leq} 1 - \eta\mu + \left( 1 + \frac{1}{4} \right)\eta^2 L^2
$$

$$
\overset{\eta = \frac{2\mu}{L^2}}{=} 1 - \frac{2\mu^2}{5L^2} + \frac{\mu^2}{5L^2} = 1 - \frac{\mu^2}{5L^2}.
$$

$\square$

**Proposition A.5.** *Take $\theta_{t+1}$ to be an approximate minima of $\ell_t(\theta)$. Suppose $T = \Pi \circ (\mathrm{Id} - \eta F)$ is a contraction, if $\ell_t(\theta_{t+1}) - \ell_t^* \to 0$ then the induced sequence $\{z_t = g(\theta_t)\}_{t \in \mathbb{N}}$ converges, $z_t \to z_*$ where $z_*$ is the unique solution to $\mathrm{VI}(\mathcal{Z}, F)$.*

*Proof.* Let $\epsilon_t = z_{t+1} - z_t^*$ be the approximation error between $z_{t+1}$ and $z_t^*$ the minimum of the surrogate $\ell_t$ (exact projected gradient step).

$$
\begin{aligned}
\frac{1}{2}\|z_{t+1} - z_*\|^2 &= \frac{1}{2}\|z_t^* - z_* + \epsilon_t\|^2 \\
&= \frac{1}{2}\|z_t^* - z_*\|^2 + \langle z_t^* - z_*, \epsilon_t \rangle + \frac{1}{2}\|\epsilon_t\|^2 \\
&\overset{\rho > 0}{\leq} \frac{1}{2}\|z_t^* - z_*\|^2 + \frac{\rho}{2}\|z_t^* - z_*\|^2 + \frac{1}{2\rho}\|\epsilon_t\|^2 + \frac{1}{2}\|\epsilon_t\|^2 \\
&\overset{\kappa \in [0,1)}{\leq} \frac{\kappa(1 + \rho)}{2}\|z_t - z_*\|^2 + (1 + \frac{1}{\rho})(\ell_t(\theta_{t+1}) - \ell_t^*).
\end{aligned}
$$

Where we use the fact that $\frac{1}{2}\|\epsilon_t\|^2 = \frac{1}{2}\|z_{t+1} - z_t^*\|^2 \leq \ell_t(\theta_{t+1}) - \ell_t^*$ from Remark A.1. Take any $\rho$ such that $\kappa(1 + \rho) < 1$ then apply Lemma 3.9 in Franci & Grammatico (2022). $\square$

**Proposition 3.3.** *There exists an $L$-smooth and $\mu$-strongly monotone $F$, and a sequence of iterates $\{z_t\}_{t \in \mathbb{N}}$ verifying the alpha descent condition with $\alpha < 1$ such that $z_t$ diverges for any $\eta$.*

*Proof.* Let us consider the simple min max example with loss

$$
\min_x \max_y \frac{1}{2}x^2 + xy - \frac{1}{2}y^2.
$$

This problem can be written as VIP where $z = (x, y)$ and the operator $F$ is linear given by the matrix

$$
F = \begin{bmatrix} 1 & 1 \\ -1 & 1 \end{bmatrix}.
$$

It is well-known that $F$ is both smooth and strongly monotone with $z_{t+1} = z_t - \eta F(z_t)$ being a contraction for small enough $\eta$ (Facchinei & Pang, 2003). Now if we consider a biased direction given by a matrix $P$, that is $z_{t+1} = z_t - P z_t$, then using the fact that $\ell_t^* = 0$ the $\alpha$-descent on the surrogate corresponds to the following bound

$$\|z_{t+1} - z_t + \eta F z_t\| = \|(\eta F - P) z_t\| \leq \alpha \|\eta F z_t\|.$$

Despite being a contraction when we follow the true gradient $F$, the above min max loss causes rotations in the dynamics that are inherent to the adversarial nature of the problem. These rotations are carefully controlled by the stepsize and strong convexity/concavity of the loss. Our counterexample simply adds a bit of rotation that ensures $\alpha < 1$ but yet is detrimental to the convergence. Mathematically, we take $P = (\mathrm{Id} - \alpha Q) \eta F$ where $Q$ is the rotation matrix

$$Q = \begin{bmatrix} \frac{1}{\sqrt{2}} & -\frac{1}{\sqrt{2}} \\ \frac{1}{\sqrt{2}} & \frac{1}{\sqrt{2}} \end{bmatrix}.$$

With this construction we are guaranteed that $\alpha < 1$ since

$$\|(\eta F - P) z_t\| = \|\alpha Q \eta F z_t\| = \alpha \eta \|F z_t\|,$$

where the last equality is due the the fact that $Q$ is an orthogonal matrix and therefore does not change the Euclidean norm of a vector.

Now taking $\alpha = 1/\sqrt{2}$ gives

$$P = \left( \mathrm{Id} - \alpha \begin{bmatrix} \frac{1}{\sqrt{2}} & -\frac{1}{\sqrt{2}} \\ \frac{1}{\sqrt{2}} & \frac{1}{\sqrt{2}} \end{bmatrix} \right) \begin{bmatrix} \eta & \eta \\ -\eta & \eta \end{bmatrix}.$$

$$= \begin{bmatrix} \frac{1}{2} & \frac{1}{2} \\ -\frac{1}{2} & \frac{1}{2} \end{bmatrix} \begin{bmatrix} \eta & \eta \\ -\eta & \eta \end{bmatrix} = \begin{bmatrix} 0 & \eta \\ -\eta & 0. \end{bmatrix}$$

We have that $z_{t+1} = z_t - P z_t = \begin{bmatrix} 1 & -\eta \\ \eta & 1 \end{bmatrix} z_t$ has an $\alpha = 1/\sqrt{2}$ but yet diverges for any $\eta > 0$ since the Eigenvalues of the linear system are $\lambda = 1 \pm i\eta$ therefore the spectral radius is strictly greater than one. Note that these dynamics are equivalent to gradient descent ascent on the bilinear loss $f(x, y) = xy$, which is known to diverge for any stepsize.

$\square$

**Theorem 3.6.** *Let $\mathcal{Z} = \mathbb{R}^n$, and Assumption (3.4) hold. If $F_\xi(x)$ is $L$-smooth and $\eta \leq \frac{1}{2(1+c)L}$ where $c \geq 2(1 + \alpha^2)$ then any trajectory $\{\theta_t\}_{t \in \mathbb{N}}$ satisfying the $\alpha$-expected descent condition guarantees:*

$$\mathbb{E}\left[\tfrac{1}{2}\|z_{t+1} - z_*\|^2\right] \leq \mathbb{E}\left[\tfrac{1}{2}\|z_t - z_*\|^2\right]\left(1 - \eta\mu + \alpha^2\right) + \eta^2(1+c)\sigma^2.$$

*Proof.* For convenience let $F_{\xi_t}(z_t) = \hat{F}_t$. Let $z_t^* = z_t - \eta \hat{F}_t$ and $p_t = z_t - z_{t+1}$, $\mathbb{E}_{\theta_{t+1}}[\cdot]$ denote the expectation over $z_{t+1} = g(\theta_{t+1})$ given $z_t$, and $\xi_t$. The expected $\alpha$-descent condition in the unconstrained case is equivalent to

$$\mathbb{E}_{\theta_{t+1}}\left[\|z_{t+1} - z_t + \eta \hat{F}_t\|^2\right] \leq \alpha^2 \eta^2 \|\hat{F}_t\|^2.$$

By concavity of the square root and the expected $\alpha$-descent condition we have $\mathbb{E}\left[\sqrt{X}\right] \leq \sqrt{\mathbb{E}[X]}$ and

$$\mathbb{E}_{\theta_{t+1}}\left[\|z_{t+1} - z_t + \eta \hat{F}_t\|\right] = \mathbb{E}_{\theta_{t+1}}\left[\sqrt{\|z_{t+1} - z_t + \eta \hat{F}_t\|^2}\right] \tag{17}$$

$$\leq \sqrt{\mathbb{E}\left[\|z_{t+1} - z_t + \eta \hat{F}_t\|^2\right]} \tag{18}$$

$$\leq \alpha \eta \|\hat{F}_t\|. \tag{19}$$

We also have the following bound

$$\mathbb{E}_{\theta_{t+1}}\left[\|z_{t+1} - z_t\|^2\right] = \mathbb{E}_{\theta_{t+1}}\left[\|z_{t+1} - z_t + \eta\hat{F}_t - \eta\hat{F}_t\|^2\right] \tag{20}$$

$$\leq \mathbb{E}_{\theta_{t+1}}\left[2\|z_{t+1} - z_t + \eta\hat{F}_t\|^2 + 2\|\eta\hat{F}_t\|^2\right] \tag{21}$$

$$\leq 2(1 + \alpha^2)\eta^2\|\hat{F}_t\|^2. \tag{22}$$

Next we expand and bound the distance $\frac{1}{2}\|z_{t+1} - z_*\|^2$.

$$\frac{1}{2}\|z_{t+1} - z_*\|^2 = \frac{1}{2}\|z_t - p_t - z_*\|^2$$

$$= \frac{1}{2}\|z_t - z_*\|^2 - \langle p_t, z_t - z_*\rangle + \frac{1}{2}\|p_t\|^2$$

$$= \frac{1}{2}\|z_t - z_*\|^2 - \eta\langle\hat{F}_t, z_t - z_*\rangle + \langle\eta\hat{F}_t - p_t, z_t - z_*\rangle + \frac{1}{2}\|p_t\|^2$$

$$\leq \frac{1}{2}\|z_t - z_*\|^2 - \eta\langle\hat{F}_t, z_t - z_*\rangle + \|\eta\hat{F}_t - p_t\|\|z_t - z_*\| + \frac{1}{2}\|p_t\|^2$$

$$= \frac{1}{2}\|z_t - z_*\|^2 - \eta\langle\hat{F}_t, z_t - z_*\rangle + \|z_{t+1} - z_t + \eta\hat{F}_t\|\|z_t - z_*\| + \frac{1}{2}\|z_{t+1} - z_t\|^2$$

Taking an expectation over $z_{t+1}$ given $z_t$, and $\xi_t$ we have

$$\mathbb{E}_{\theta_{t+1}}\left[\frac{1}{2}\|z_{t+1} - z_*\|^2\right] \overset{(19,22)}{\leq} \frac{1}{2}\|z_t - z_*\|^2 - \eta\langle\hat{F}_t, z_t - z_*\rangle + \eta\alpha\|\hat{F}_t\|\|z_t - z_*\| + \frac{2(1 + \alpha^2)\eta^2}{2}\|\hat{F}_t\|^2$$

$$\leq \frac{1}{2}\|z_t - z_*\|^2 - \eta\langle\hat{F}_t, z_t - z_*\rangle + \eta\alpha\|\hat{F}_t\|\|z_t - z_*\| + \frac{c\eta^2}{2}\|\hat{F}_t\|^2$$

$$\leq \frac{1}{2}\|z_t - z_*\|^2 - \eta\langle\hat{F}_t, z_t - z_*\rangle + \frac{\alpha^2}{2}\|z_t - z_*\|^2 + \frac{\eta^2(1 + c)}{2}\|\hat{F}_t\|^2$$

where in the inequality we used (19, 22) and in the second inequality $c \geq 2(1 + \alpha^2)$. Taking an expectation given information up to time $t$ and using the expected co-coercivity assumption gives,

$$\mathbb{E}_t\left[\frac{1}{2}\|z_{t+1} - z_*\|^2\right] \leq \frac{1}{2}\|z_t - z_*\|^2 - \eta\langle F_t, z_t - z_*\rangle + \frac{\alpha^2}{2}\|z_t - z_*\|^2 + \frac{\eta^2(1 + c)}{2}\mathbb{E}_t\left[\|\hat{F}_t\|^2\right]$$

$$\overset{(3.4)}{\leq} \frac{1}{2}\|z_t - z_*\|^2 - \eta\langle F_t, z_t - z_*\rangle + \frac{\alpha^2}{2}\|z_t - z_*\|^2 + \eta^2(1 + c)L\langle F_t, z_t - z_*\rangle + \eta^2(1 + c)\sigma^2$$

$$= \frac{1}{2}\|z_t - z_*\|^2 - \eta\left(1 - \eta(1 + c)L\right)\langle F_t, z_t - z_*\rangle + \frac{\alpha^2}{2}\|z_t - z_*\|^2 + \eta^2(1 + c)\sigma^2$$

$$\overset{\eta(1+c)L<1}{\leq} \frac{1}{2}\|z_t - z_*\|^2 - \eta\mu\left(1 - \eta(1 + c)L\right)\|z_t - z_*\|^2 + \frac{\alpha^2}{2}\|z_t - z_*\|^2 + \eta^2(1 + c)\sigma^2$$

If we take $\eta < \frac{1}{2(1+c)L}$ then we have the following recursion

$$\mathbb{E}_t\left[\frac{1}{2}\|z_{t+1} - z_*\|^2\right] \leq \frac{1}{2}\|z_t - z_*\|^2\left(1 - \eta\mu + \alpha^2\right) + \eta^2(1 + c)\sigma^2.$$

Taking an expectation over $z_t$ gives the result. □

## B   NON-LINEAR LEAST-SQUARES

**Lemma B.1.** *Let $f(\theta) = \frac{1}{2}\|r(\theta)\|^2$ where $r : \mathbb{R}^d \to \mathbb{R}^n$, $r(\theta) = (r_i(\theta), \cdots, r_n(\theta))^\top$. Suppose $r_i(\theta)$ is $\beta$-smooth and $Dr(\theta)^\top$ has singular values that are globally upper bounded by $\sigma_{\max} > 0$. If $r$ is bounded, then $f$ is $L$-smooth: there exists $L \geq 0$ such that $\|\nabla f(\theta) - \nabla f(\theta')\| \leq L\|\theta - \theta'\|$ for any $\theta, \theta' \in \mathbb{R}^d$.*

*Proof.* Since each $r_i$ is $\beta$-smooth we have that

$$\|Dr(\theta)^\top - Dr(\theta')^\top\|_{1,2} = \sup\{\|(Dr(\theta)^\top - Dr(\theta')^\top)v\|_1 : v \in \mathbb{R}^n, \|v\|_2 \leq 1\}$$
$$= \max_i \|\nabla g_i(\theta) - \nabla g_i(\theta')\| \leq L\|\theta - \theta'\|.$$

$$\|\nabla f(\theta) - \nabla f(\theta')\| = \|Dr(\theta)^\top r(\theta) - Dr(\theta')^\top r(\theta')\|$$
$$= \|Dr(\theta)^\top r(\theta) - Dr(\theta)^\top r(\theta') + Dr(\theta)^\top r(\theta') - Dr(\theta')^\top r(\theta')\|$$
$$\leq \|Dr(\theta)^\top r(\theta) - Dr(\theta)^\top r(\theta')\| + \|Dr(\theta)^\top r(\theta') - Dr(\theta')^\top r(\theta')\|$$
$$\leq \sigma_{\max}\|r(\theta) - r(\theta')\| + \|Dr(\theta)^\top - Dr(\theta')^\top\|_{1,2}\|r(\theta')\|_1$$
$$\leq \sigma_{\max}^2\|\theta - \theta'\| + \beta C\|\theta - \theta'\|.$$

Where the last inequality follows from $r$ being $\sigma_{\max}$ Lipschitz since $\|r(\theta) - r(\theta')\| \leq \|Dr(\theta)\|\|\theta - \theta'\|$ by the mean value inequality (Hörmander, 2007) and by the fact that $\|Dr(\theta)^\top\| \leq \sigma_{\max}$. Since $r$ is bounded it follows that there exists $\|r(\theta)\|_1 \leq C$ for all $\theta$. $\qquad\square$

**Definition B.2** (Polyak (1964); Łojasiewicz (1963)). *A function $f : \mathbb{R}^n \to \mathbb{R}$ satisfies the PL condition if there exists $\mu > 0$ such that for all $x \in \mathbb{R}^n$*

$$\|\nabla f(x)\|^2 \geq 2\mu(f(x) - f^*), \tag{23}$$

*where $f^* = \inf_{x \in \mathbb{R}^n} f(x)$.*

**Proposition 4.1.** *Assume $f$ satisfies the $\mu$-PL condition, and let $\sigma_{\min}$ be a lower bound on the singular values of $Dg(\theta)^\top$. Then, we have $f \circ g$ is $\mu\sigma_{\min}^2$-PL.*

*Proof.* We have that

$$\|\nabla f \circ g(\theta)\|^2 = \|Dg(\theta)^\top \nabla_{z=g(\theta)} f(z)\|^2$$
$$= \nabla f(z)^\top D(g(\theta))D(g(\theta))^\top \nabla f(z)$$
$$\geq \sigma_{\min}^2\|\nabla f(z)\|^2$$
$$\geq \sigma_{\min}^2\mu(f(z) - \inf_{z'} f(z'))$$
$$\geq \sigma_{\min}^2\mu(f(z) - \min_{z' \in \mathcal{Z}} f(z'))$$
$$= \sigma_{\min}^2\mu(f(g(\theta)) - \inf_{\theta'} f(g(\theta'))).$$

$\qquad\square$

**Proposition B.4.** *If $g : \mathbb{R}^d \to \mathbb{R}^n$ $g(\theta) = (g_1(\theta), \cdots, g_n(\theta))^\top$ is differentiable where $g_i$ is $\beta$-smooth and $Dg(\theta)^\top$ has globally lower and upper bounded singular values, then $\{g(\theta) : \theta \in \mathbb{R}^d\}$ is unbounded.*

*Proof.* We prove by contradiction. Suppose $\{g(\theta) : \theta \in \mathbb{R}^d\}$ is bounded and denote $\mathcal{Z}$ as its closure which is also bounded. Then there exists $v \in \mathbb{R}^n \notin \mathcal{Z}$. Consider the function $f(\theta) = \frac{1}{2}\|g(\theta) - v\|^2$. Taking $r(\theta) = g(\theta) - v$, we have $Dr = Dg$, and $r_i$ are still $\beta$-smooth, therefore by Lemma B.1 $f$ is $L$-smooth. Since $f$ is continuous and $\mathcal{Z}$ is compact, its minimum is attained, and thus there exists $z_* \in \mathcal{Z}$ such that

$$\frac{1}{2}\|z_* - v\|^2 = \inf_{\theta \in \mathbb{R}^d} f(\theta).$$

Moreover, there exists a sequence $\{\theta_t\}_{t \in \mathbb{N}}$ such that $g(\theta_t) \to z_*$ since $\mathcal{Z}$ is closed. By the smoothness of $f$ it also follows that

$$\frac{\|\nabla f(\theta_t)\|^2}{2L} \leq f(\theta_t) - \inf_{\theta \in \mathbb{R}^d} f(\theta).$$

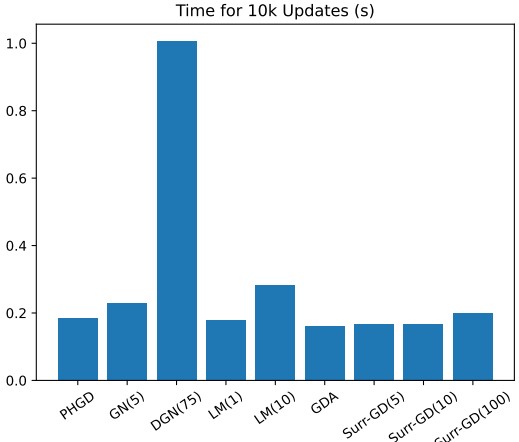

Figure 5: Time in seconds for performing 10,000 iterations of each method. The number in paren-thesis correspond to number of inner steps taken. As a special case we have PHGD and GDA are equivalent to GN(1) and Surr-GD(1) respectively.

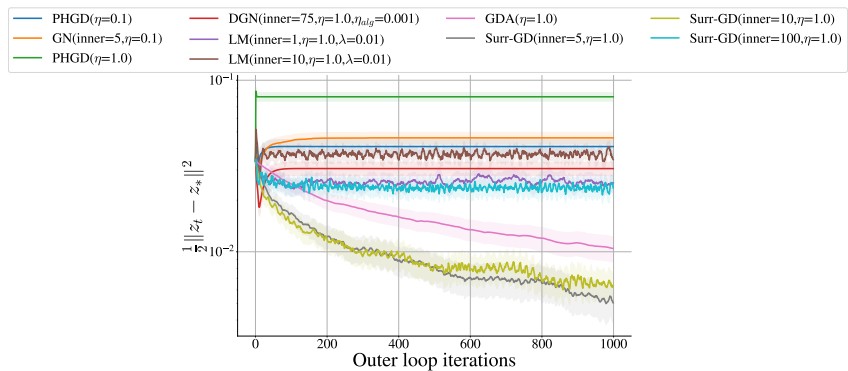

Figure 6: Average convergence for various surrogate methods from Section 4 in the hidden matching pennies game. The average was computed over 1,000 trajectories each with randomly sampled parameters $\alpha_j^i$ and $\theta_0$.

Since singular values of $Dg(\theta)^\top$ are lower-bounded, there exists $\sigma_{\min} > 0$ such that $\|Dg(\theta)^\top v\| \geq \sigma_{\min}\|v\|$. Therefore we have

$$0 = \lim_t f(\theta_t) - \inf_{\theta \in \mathbb{R}^d} f(\theta) \geq \lim_t \frac{\|\nabla f(\theta_t)\|^2}{2L}$$

$$= \lim_t \frac{\|Dg(\theta_t)^\top(g(\theta_t) - v)\|^2}{2L} \geq \lim_t \frac{\sigma_{\min}^2}{2L}\|g(\theta_t) - v\|^2,$$

which implies that $z_* = \lim_t g(\theta_t) = v$ and that $v$ is indeed within $\mathcal{Z}$ a contradiction!

□

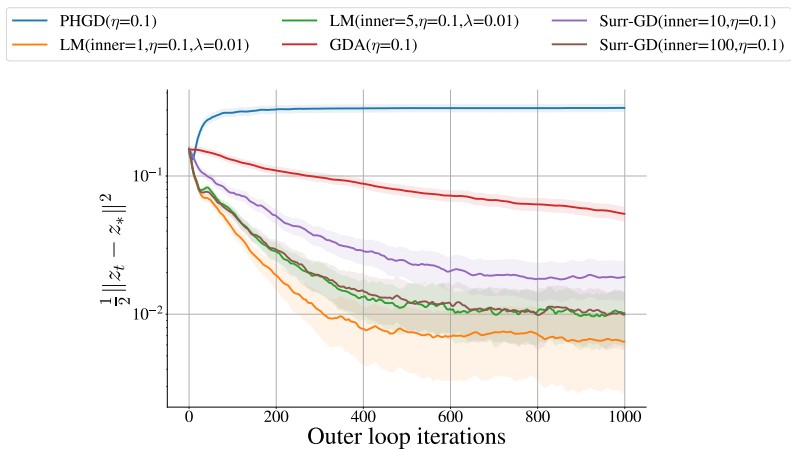

Figure 7: Average convergence for various surrogate methods from Section 4 in the hidden rock-paper-scissors game. The average was computed over 1,000 trajectories each with randomly sampled parameters $A_j^i$ and $\theta_0$.

## C    MIN MAX EXPERIMENTS

### C.1    HIDDEN MATCHING PENNIES

In the hidden matching pennies game each player $i$ has the parameterization $h^i(\theta) = \text{sigmoid}(\alpha_2^i \, \text{CELU}(\alpha_1^i \theta))$. The parameters for each player are:

$$\alpha_1^1 = 0.5, \text{ and } \alpha_2^1 = 1 \text{ for player 1,}$$
$$\alpha_1^2 = 0.7, \text{ and } \alpha_2^2 = 1 \text{ for player 2.}$$

The parameters $\alpha_1^i, \alpha_2^i$, were chosen to approximately replicate the trajectory of PHGD presented in Sakos et al. (2024, Figure 4). The initial parameters $\theta_1$ were selected to be the same as those selected by Sakos et al. (2024), $\theta_1 = (\theta_1^1, \theta_1^2) = (1.25, 2.25)$.

In Figure 5 we report the runtime for different surrogate algorithms and number of inner steps. In Figure 6, the average distance across 100 random initializations is observed. $\theta_0$ was randomly sampled from two independent Gaussians with standard deviation of 4. Similarly, each $\alpha_j^i$ was randomly sampled from a uniform random variable with support $[-1, 1]$.

### C.2    HIDDEN ROCK-PAPER-SCISSORS

Each player i's mixed strategy in the hidden rock-paper-scissors game is parameterized by the function $h^i(\theta) = \text{softmax}(A_2^i \, \text{CELU}(A_1^i \theta^i))$, with $\theta^i \in \mathbb{R}^5$. The min-max objective is $f(z^1, z^2) = \langle z^1, A z^2 \rangle + \frac{0.2}{2} \|z - z_*\|^2$, where $z = (z^1, z^2)$ and $z_*$ is the equilibrium of rock-paper-scissors (sample rock, paper, and scissors with equal probability) (Sakos et al., 2024). In Figure 7 we observe the average convergence across 100 random initializations of $\theta_1 \in \mathbb{R}^5$ and matrixes $A_j^i$ for various surrogate based methods. $\theta_1$ was sampled from a 5-dimensional isotropic Gaussian while each element of $A_j^i$ was sampled from a uniform random variable with support $[-1, 1]$.

## D    POLICY EVALUATION AND PROJECTED BELLMAN ERROR

In this section we provide some background on minimizing projected Bellman error (PBE) and the stochastic methods proposed by Bertsekas (2009). For a given MDP with $n$ states $(s^1, \cdots, s^n)$, and policy $\pi$, we denote the expected reward $r_\pi \in \mathbb{R}^n$, discount factor $\gamma \in (0, 1)$, and state to state probability transition matrix $P_\pi \in \mathbb{R}^{n \times n}$. The associated linear Bellman operator is $T_\pi(z) = r_\pi + \gamma P_\pi z$ and is contraction Bertsekas (2012). The value vector $v_\pi \in \mathbb{R}^n$ is the unique fixed point of $T_\pi$, with i$^{\text{th}}$ component $v^i$ equals to the discounted return starting from state

$s^i$, $\mathbb{E}_{\tau=(s_1,s_2,\cdots)}\left[\sum_{t=1}^{\infty}\gamma^t r_\pi(s_t)|s_1=s^i\right]$, where we use $r_\pi(s)$ to mean $r_\pi^i$ such that $s^i=s$, and the expectation is over possible trajectories $\tau=(s_1,s_2,\cdots)$.

Under function approximation it may not be possible to find a fixed point of $T_\pi$ and find the true value function. Instead we can consider the projected fixed point,

$$z_* = \Pi_\Xi(T_\pi(z_*)). \tag{24}$$

Where the projection is on to the set of realizable value functions $\mathcal{Z}=\{v_\theta : \theta \in \mathbb{R}^d\}$. We use the notation $v_\theta$ in place of $g(\theta)$ to be consistent with the RL literature, the prediction for state $s^i$ is $v_\theta(s^i)=g(\theta)^i$. $\Xi$ is the diagonal matrix with entries corresponding to the stationary distribution $\xi=(\xi^1,\cdots,\xi^n)$ of $P_\pi$.

As pointed out by Bertsekas (2009), finding the fixed point (24) is equivalent to solving a VI of the form (1) with constraint $\mathcal{Z}$ and operator

$$F(z) = \Xi(z - T_\pi(z)). \tag{25}$$

Additionally, Bertsekas (2009) showed that $F$ is both smooth and strongly monotone and therefore satisfies our hidden structure Assumption 3.1.

This fixed point can then be equivalently mapped to the minimum of the projected Bellman error with respect to the parameters, i.e. a TD fixed point. The fixed point $z_* = v_{\theta_*}$ corresponds to finding parameters $\theta_*$ such that

$$\theta_* \in \arg\min_\theta\left[BE(\theta,\theta_*)=\frac{1}{2}\sum_{i=1}^n\xi^i(v_\theta(s^i)-(r_\pi(s^i)-\gamma\mathbb{E}_{s'\sim P_\pi}\left[v_{\theta_*}(s')|s^i\right])^2\right]. \tag{26}$$

Note this is not a scalar minimization problem since the loss is not directly minimized through the next state prediction, doing so would result in minimizing the Bellman error, $\min_\theta BE(\theta,\theta)$ and not PBE. The fixed points of these objectives need not coincide even with linear function approximation (Sutton & Barto, 2018).

Using the function $BE$ and the stationary condition (26) we can define the following natural gap function

$$BE_{\text{gap}}(\theta) = BE(\theta,\theta) - \inf_{\theta'}BE(\theta',\theta). \tag{27}$$

Notice that $BE_{\text{gap}}(\theta) \geq 0$ for all $\theta$ and is zero if and only if $\theta$ is a fixed point of (26).

Since $\Pi_\Xi \circ T_\pi$ is known to be a contraction (Bertsekas, 2012) but is expensive to compute exactly, we can devise a surrogate loss that approximates it. The operator $\Pi_\Xi \circ T_\pi$ is equivalent to the scaled projected gradient method $z_{t+1}=\Pi(z_t-\Xi^{-1}F(z_t))$, where $z_t=v_{\theta_t}$, and is minimum of the surrogate loss

$$\ell_t(\theta) = \frac{1}{2}\|v_\theta - \left(v_{\theta_t}-\Xi^{-1}F(v_{\theta_t})\right)\|_\Xi^2 \tag{28}$$

$$= \frac{1}{2}\|v_\theta - T_\pi(v_{\theta_t})\|_\Xi^2. \tag{29}$$

Note that if $\Xi = \text{Id}$ than this is the same surrogate loss (4) with $\eta = 1$.

### D.1 THE LINEAR CASE

In this section we assume that predicted values are linear, $g(\theta) = \Phi\theta$, with some feature matrix

$$\Phi = \begin{bmatrix} \phi^\top(s^1) \\ \vdots \\ \phi^\top(s^n) \end{bmatrix} \in \mathbb{R}^{n\times d}.$$

In this case we have that the operator $F$ is also linear in $\theta$

$$F(z) = F(\Phi\theta) = \Xi(\Phi\theta - \gamma P_\pi\Phi\theta - r_\pi) = \Xi(\Phi - \gamma P_\pi\Phi)\theta - \Xi r_\pi.$$

Since $g$ linear, there exists a preconditioning scheme that can generate a sequence of parameters $\{\theta_t\}_{t \in \mathbb{N}}$ such that $\{z_{t+1} = \Phi\theta_{t+1} = \Pi_\Xi(T_\pi(z_t))\}$ and therefore is guaranteed to converge linearly. This preconditioning scheme is simply the minimizer of the weighted least-squares surrogate loss $\ell_t(\theta) = {}^{1}\!/\!{}_{2}\|\Phi\theta - T_\pi(\theta_t)\|_\Xi^2$,

$$\theta_{t+1} = \theta_t - (\Phi^\top \Xi \Phi)^{-1} \Phi^\top F(z_t).$$

If there is no unique minimizer then the pseudo-inverse $(\Phi^\top \Xi \Phi)^\dagger$ can be used. To approximate the iteration Bertsekas (2009) suggests using estimators that depend only on the history states $(s_1, s_2, \cdots, s_t, s_{t+1})$ and rewards $(r_1, \cdots, r_t)$ to approximate the conditioning matrix $\Phi^\top \Xi \Phi$ and gradient $\Phi^\top F(z_t)$. The following estimators are proposed:

- $\hat{D}_t = \frac{1}{t} \sum_{i=1}^t \phi(s_i) \phi^\top(s_i)$

- $\hat{C}_t = \frac{1}{t} \sum_{i=1}^t \phi(s_i)(\phi(s_i) - \gamma\phi(s_{i+1}))^\top$

- $\hat{r}_t = \frac{1}{t} \sum_{i=1}^t \phi(s_i) r_i.$

If the chain is fully mixed then the sampled states are drawn according to $\xi$ making the above estimators unbiased:

$$\Phi^\top \Xi \Phi = \sum_i^n \xi^i \phi(s^i) \phi(s^i)^\top = \mathbb{E}\left[\hat{D}_t\right],$$

$$\Phi^\top F(z) = \sum_i \xi^i \phi(s^i)(\phi(s^i)^\top \theta - \sum_{s'} (P_\pi)_{s^i,s} \phi(s)^\top \theta - r_\pi(s^i)) = \mathbb{E}\left[\hat{C}_t \theta - \hat{r}_t\right].$$

Where $(P_\pi)_{s,s'}$ is the transition probability to state $s'$ when in state $s$. Therefore the iteration

$$\theta_{t+1} = \theta_t - (\hat{D}_t)^{-1}(\hat{C}_t \theta_t - \hat{r}_t),$$

eventually converges to exact deterministic iteration. Interestingly, the stochastic version can be shown to be the minimmum of the surrogate

$$\tilde{\ell}_t(\theta) = \frac{1}{2}\|\Phi\theta - \hat{T}_\pi(z_t)\|_{\hat{\Xi}}^2.$$

Where $\hat{T}_\pi(z) = \bar{r} + \gamma \bar{P}_\pi(z)$, with $\Xi$, $r$, and $P_\pi$ being estimated with the empirical distribution. More precisely, denoting the number of times that a state $s$ is visited in a trajectory of length $t$ as $n(s)$, and the number of times $s'$ is visited after $s$ as $n(s, s')$, then the empirical distribution over states is $\hat{\xi} = ({}^{n(s^1)}\!/\!{}_t, \cdots, {}^{n(s^n)}\!/\!{}_t)$ and $\hat{\Xi}$ is the diagonal matrix with diagonal $\hat{\xi}$. Similarly, the transition matrix is estimated by $(\bar{P}_\pi)_{s,s'} = n(s, s')/n(s)$ if $n(s) > 0$, and arbitrary otherwise. For convenience, we write $\hat{\xi}(s^i) = \hat{\xi}^i$.

**Proposition D.1.** *The gradient of the surrogate loss* $\tilde{\ell}_t(\theta) = \frac{1}{2}\|\Phi\theta - \hat{T}_\pi(z_t)\|_{\hat{\Xi}}^2$ *is*

$$\nabla\tilde{\ell}_t(\theta) = \hat{C}_t \theta_t - \hat{r} + \hat{D}_t(\theta - \theta_t).$$

*If* $\hat{D}_t$ *is invertible then its minimum is*

$$\theta_{t+1} = \theta_t - (\hat{D}_t)^{-1}(\hat{C}_t \theta_t - \hat{r}_t).$$

*Proof.* By definition of $\hat{\Xi}$, we have that $\hat{D}_t = \frac{1}{t}\sum_i \phi(s_i)\phi(s_i)^\top = \Phi^\top\hat{\Xi}\Phi$. Similarly, we have that $\hat{C}_t = \frac{1}{t}\sum_{i=1}^t \phi(s_i)(\phi(s_i) - \gamma\phi(s_{i+1}))^\top = \Phi^\top\hat{\Xi}\Phi - \gamma\frac{1}{t}\sum_{i=1}^t \phi(s_i)\phi(s_{i+1})^\top$. Where

$$\frac{1}{t}\sum_{i=1}^t \phi(s_i)\phi(s_{i+1})^\top = \frac{1}{t}\sum_{s,s'} n(s,s')\phi(s)\phi(s')^\top \tag{30}$$

$$= \frac{1}{t}\sum_s \phi(s)\sum_{s'} n(s,s')\phi(s')^\top \tag{31}$$

$$= \frac{1}{t}\sum_s n(s)\phi(s)\sum_{s'} \frac{n(s,s')}{n(s)}\phi(s')^\top \tag{32}$$

$$= \frac{1}{t}\sum_s n(s)\phi(s)\sum_{s'} (\bar{P}_\pi)_{s,s'}\phi(s')^\top \tag{33}$$

$$= \sum_s \hat{\xi}(s)\phi(s)\sum_{s'} (\bar{P}_\pi)_{s,s'}\phi(s')^\top \tag{34}$$

$$= \Phi^\top\hat{\Xi}\bar{P}_\pi\Phi. \tag{35}$$

Therefore $\hat{C}_t = \Phi^\top\hat{\Xi}(\Phi - \gamma\bar{P}_\pi\Phi)$. Similarly, $\hat{r}_t = \frac{1}{t}\sum_{i=1}^t \phi(s_i)r_i = \Phi^\top\Xi\bar{r}_\pi$.

Now consider the weighted least squares loss

$$\tilde{\ell}_t(\theta) = \frac{1}{2}\|\Phi\theta - \hat{T}_\pi(\Phi\theta_t)\|_{\hat{\Xi}}^2.$$

The gradient of $\ell_t$ is

$$\nabla\ell_t(\theta) = \Phi^\top\hat{\Xi}(\Phi\theta - \hat{T}_\pi(\Phi\theta_t)) = \Phi^\top\hat{\Xi}(\Phi\theta - \gamma\bar{P}_\pi\Phi\theta_t - \bar{r}_\pi) \tag{36}$$

$$= \Phi^\top\hat{\Xi}(\Phi\theta - \Phi\theta_t + \Phi\theta_t - \gamma\bar{P}_\pi\Phi\theta_t - \bar{r}_\pi) \tag{37}$$

$$= \Phi^\top\hat{\Xi}(\Phi\theta - \Phi\theta_t) + \Phi^\top\hat{\Xi}(\Phi\theta_t - \gamma\bar{P}_\pi\Phi\theta_t - \bar{r}_\pi) \tag{38}$$

$$= \hat{D}_t(\theta - \theta_t) + \hat{C}_t\theta_t - \hat{r}_t \tag{39}$$

Setting the gradient to zero gives the minimum of $\tilde{\ell}_t$. □

## D.2 THE NONLINEAR CASE

In this section, we detail the surrogate algorithms used for the nonlinear setting, as outlined in Algorithms 2, 3 and 4. As a complement to Section 5.2, we compare the convergence properties of TD(0) with the surrogate methods under a an offline, full-batch setting, where the training data is pre-collected from the reference policy. This experiment provides insight on two important fronts: how well can the methods solve the hidden monotone VI problem, and how the performance of the different algorithms are affected when there is no environment interaction.

In our deep learning/nonlinear setting, the state space is too large to verify if we have solved the VI (25) by the $BE_{\text{gap}}$(27) or other metrics. To test the effectiveness of our methods at solving this VI and minimizing PBE we consider an empirical approximation given a fixed batch of data (i.e. offline). In this setting, $n$ tuples of state, reward and next state are collected $\{(s_t, r_t, s_{t+1})\}_{t=1}^n$ from a fixed reference policy $\pi$. Using these observations we construct an empirical estimate of $BE$ (26),

$$\widehat{BE}(\theta, \theta') = \frac{1}{n}\sum_{t=1}^n (v_\theta(s_t) - (r_t - \gamma v_{\theta'}(s_{t+1})))^2. \tag{40}$$

We can then define the following fixed point problem: find $\theta_*$ such that

$$\theta_* \in \arg\min_\theta \widehat{BE}(\theta, \theta_*). \tag{41}$$

Given this new empirical approximation of PBE we can define a gap analogous to (27),

$$\widehat{BE}_{\text{gap}}(\theta) = \widehat{BE}(\theta, \theta) - \inf_{\theta'} \widehat{BE}(\theta', \theta). \tag{42}$$

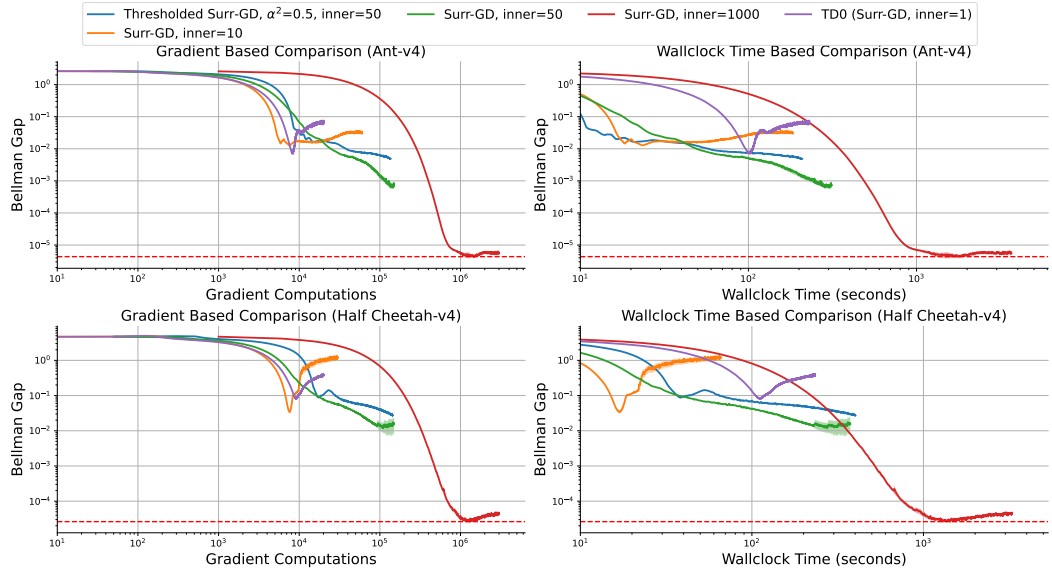

Figure 8: Comparison of Bellman gap during training of TD(0) and surrogate methods in the offline full-batch setting with nonlinear function approximation in Ant (top) and HalfCheetah (bottom) environments, measured by gradient computations (left) and wallclock time (right). The average Bellman gap (42) across 20 runs along with 95% confidence intervals are computed from a fixed training set. The red dashed line represents the lowest Bellman gap achieved by any of the algorithms.

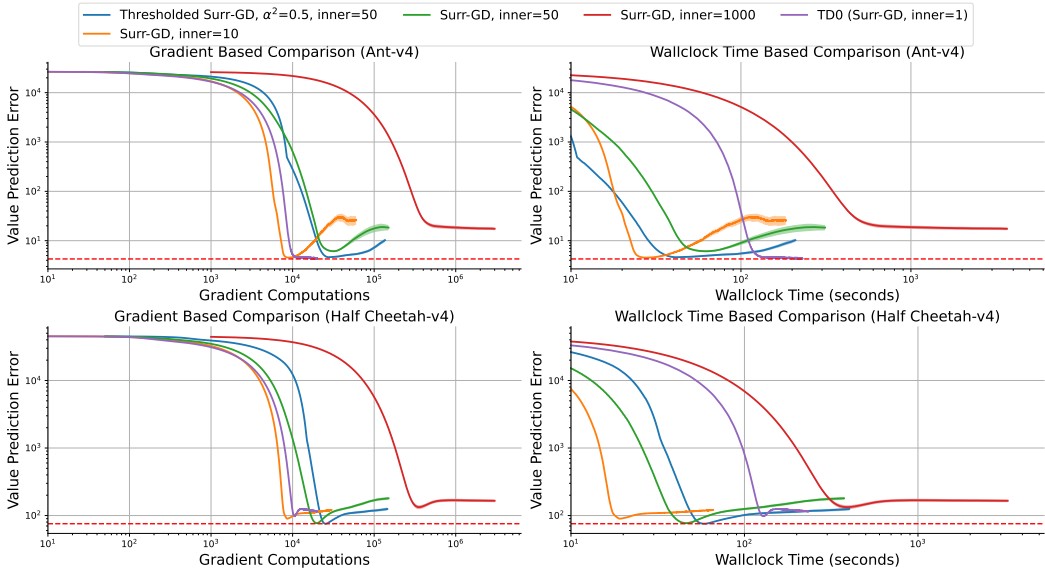

Figure 9: Comparison of average performance of TD(0) and surrogate methods in minimizing the value prediction error in the offline full-batch setting with nonlinear function approximation in Ant (top) and HalfCheetah (bottom) environments, measured by gradient computations (left) and wallclock time (right). The average value prediction error across 20 runs along with 95% confidence intervals are computed from a fixed test set. The red dashed line represents the lowest value prediction error achieved by any of the algorithms.

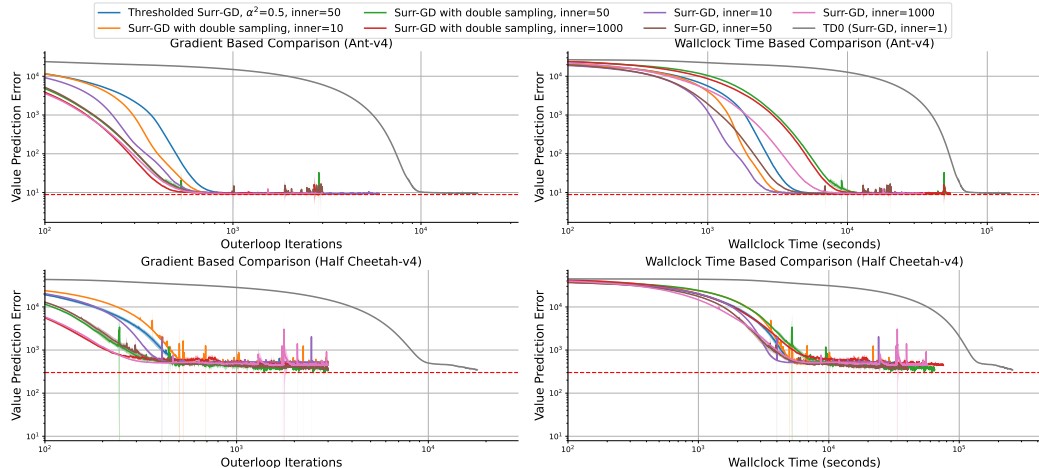

Figure 10: Comparison of average performance of surrogate methods in minimizing the value prediction error for RL tasks with a 16-layer MLP in Ant as measured by outer loop iterations (left) and wallclock time (right). The average value prediction error across 20 runs along with 95% confidence intervals are computed from a fixed test set. The red dashed line represents the lowest value prediction error achieved by any of the algorithms.

In practice we approximate the infimum in $\widehat{BE}$ by 500 steps of AdamW, with a learning rate of $1 \times 10^{-4}$ and exponential annealing at a decay rate of $\gamma = 0.995$. In the offline setting we consider deterministic algorithms like in the min-max experiments of Section 5.1, and therefore we do not consider the double sampling algorithm (Algorithm 3) but only full-batch versions of Algorithm 2 and Algorithm 4. We compare different algorithms with $\widehat{BE}_{\text{gap}}$ (42), giving a measure of how much the fixed point condition (41) is violated. Given enough data this fixed point is a proxy for the fixed point (26).

Since no environment interaction is required during training, we compare the performance of the methods in terms of gradient computations and wall-clock time, with the former being the dominant factor in computational cost, unlike the setting in Section 5.2 where environment interaction plays a larger role.

As shown in Figure 8, increasing the number of inner loop steps significantly reduces the Bellman gap, suggesting that the model approaches the fixed point more effectively. Notably, the surrogate method with 10 inner steps is more efficient than TD(0) in terms of gradient computations, while both inner steps = 10 and 50 outperform TD(0) in wall-clock time.

As presented in Figure 9, we also evaluate the value prediction error on a test set to further assess the performance of the algorithms in the full batch and deterministic setting. In all cases, at least one variant of the surrogate method converges faster than TD(0), both in terms of gradient computations and wall-clock time. This demonstrates that, with an appropriate choice of inner loop steps and alpha threshold, the surrogate method not only outperforms TD(0) in terms of stability but also in computational efficiency.

### D.2.1 LARGER NETWORK

In Figure 10 we report the average performance of surrogate methods using a 16 layer MLP. We observe similar runtimes and performance to the 2 layer MLP used in Section 4.

---

**Algorithm 2:** Surrogate Method with Inner Loop

---

**Input:** Reference policy $\pi$, initial value function parameters $\theta_0$, length of trajectories $N$, discount factor $\gamma$, learning rate $\eta$, number of inner steps $M$, number of outer loop iterations T, stop gradient function $\text{sg}(\cdot)$

**for** $t = 0$ **to** $T$ **do**

    Collect a batch of trajectories $\{(s_i, a_i, r_i, s_i')\}_{i=1}^N$ from $\pi$

    **for** *each $(s_i, r_i, s_i')$ in the batch* **do**

        Compute TD target: $y_i = r_i + \gamma\text{sg}(V_{\theta_t}(s_i'))$

    Initialize inner loop $\theta$: $\theta_t^{(0)} = \theta_t$

    **for** $m = 0$ **to** $M - 1$ **do**

        **for** *each $(s_i, r_i, s_i')$ in the batch* **do**

            Compute TD error: $\delta_i^{(m)} = V_{\theta_t^{(m)}}(s_i) - y_i$

        Compute mean squared error: $\frac{1}{N}\sum_{i=1}^N (\delta_i^{(m)})^2$

        Update inner loop $\theta$: $\theta_t^{(m+1)} \leftarrow \theta_t^{(m)} - \eta\frac{1}{N}\sum_{i=1}^N \delta_i^{(m)}\nabla_\theta V_{\theta_t^{(m)}}(s_i)$

    Update outer loop $\theta$: $\theta_{t+1} = \theta_t^{(M)}$

---

**Algorithm 3:** Surrogate Method with Double Sampling

---

**Input:** Reference policy $\pi$, initial value function parameters $\theta_0$, length of trajectories $N$, buffer size $K$, discount factor $\gamma$, learning rate $\eta$, number of inner steps $M$, number of outer loop iterations T, stop gradient function $\text{sg}(\cdot)$

Collect $K$ batches of trajectories $\{(s_i, a_i, r_i, s_i')\}_{i=1}^N$ from $\pi$ to construct a buffer

**for** $t = 0$ **to** $T$ **do**

    Collect a batch of trajectories $\{(s_i, a_i, r_i, s_i')\}_{i=1}^N$ from $\pi$

    **for** *each $(s_i, r_i, s_i')$ in the batch* **do**

        Compute TD target: $y_i = r_i + \gamma\text{sg}(V_{\theta_t}(s_i'))$

        Compute TD error: $F(V_{\theta_t}(s_i)) = \text{sg}(V_{\theta_t}(s_i)) - y_i$

    Initialize inner loop $\theta$: $\theta_t^{(0)} = \theta_t$

    **for** $m = 0$ **to** $M - 1$ **do**

        Sample $m \mod K$-th batch $\{(s_j, a_j, r_j, s_j')\}_{j=1}^N$ from the buffer

        **for** *each $(s_i, r_i, s_i')$ in the new batch* **do**

            Compute linearization term $l_1(\theta_t^{(m)}) = \langle F(V_{\theta_t}(s_i)), V_{\theta_t^{(m)}}(s_i) - \text{sg}(V_{\theta_t}(s_i))\rangle$

        **for** *each $(s_j, r_j, s_j')$ in the buffer batch* **do**

            Compute regularization term $l_2(\theta_t^{(m)}) = \frac{1}{2}\|V_{\theta_t^{(m)}}(s_j) - \text{sg}(V_{\theta_t}(s_j))\|_2^2$

        Construct the surrogate loss function $L(\theta_t^{(m)}) = \frac{1}{N}(l_1(\theta_t^{(m)}) + l_2(\theta_t^{(m)}))$

        Update inner loop $\theta_t^{(m)}$ by minimizing the surrogate loss:

        $\theta_t^{(m+1)} \leftarrow \theta_t^{(m)} - \eta\nabla_\theta L(\theta_t^{(m)})$

    Update outer loop $\theta$: $\theta_{t+1} = \theta_t^{(M)}$

---

---

**Algorithm 4:** Thresholded Surrogate Method

---

**Input:** Reference policy $\pi$, initial value function parameters $\theta_0$, length of trajectories $N$, discount factor $\gamma$, learning rate $\eta$, number of inner steps $M$, number of outer loop iterations T, stop gradient function $\mathrm{sg}(\cdot)$, $\alpha^2 \in (0, 1)$

**for** $t = 0$ **to** $T$ **do**

    Collect a batch of trajectories $\{(s_i, a_i, r_i, s_i')\}_{i=1}^N$ from $\pi$

    **for** *each* $(s_i, r_i, s_i')$ *in the batch* **do**

        | Compute TD target: $y_i = r_i + \gamma \mathrm{sg}(V_{\theta_t}(s_i'))$

    Initialize inner loop $\theta$: $\theta_t^{(0)} = \theta_t$

    Compute initial surrogate loss: $l_t(\theta_t) = \frac{1}{N}\sum_{i=1}^N \|V_{\theta_t}(s_i) - y_i\|_2^2$

    **while** $l_t(\theta_t^{(m)})/l_t(\theta_t) \geq \alpha^2$ *and* $m \leq M$ **do**

        **for** *each* $(s_i, r_i, s_i')$ *in the batch* **do**

            | Compute TD error: $\delta_i^{(m)} = V_{\theta_t^{(m)}}(s_i) - y_i$

        Compute mean squared error: $\frac{1}{N}\sum_{i=1}^N (\delta_i^{(m)})^2$

        Update inner loop $\theta$: $\theta_t^{(m+1)} \leftarrow \theta_t^{(m)} - \eta\frac{1}{N}\sum_{i=1}^N \delta_i^{(m)}\nabla_\theta V_{\theta_t^{(m)}}(s_i)$

        Update $m : m \leftarrow m + 1$

    Update outer loop $\theta$: $\theta_{t+1} = \theta_t^{(M)}$

---

# E  APPLICATIONS TO PERFORMATIVE PREDICTION

Minimizing Projected Bellman error can be viewed as a special case of finding a stable point in performative prediction (Perdomo et al., 2020). In performative prediction or more generally optimization under decision dependent distributions (Drusvyatskiy & Xiao, 2023), the model predictions $z = g(\theta)$ influence the targets $y$. With moving targets it is of interest to find model predictions that are performatively stable; such models are optimal under the shifting targets and are stable under repeated applications of empirical risk minimization.

For simplicity, we stay within the finite dimensional case where a model's prediction function is given by the $n$-dimensional vector, $z = g(\theta) \in \mathbb{R}^n$. Where each prediction is $z^i = g^i(\theta) = h(x_i, \theta)$, for some feature vector $x_i$ and fixed model architecture $h$. Given an error function $\ell(z, y)$, quantifying error between predictions $z \in \mathbb{R}^n$ and targets $y \in \mathbb{R}^n$, the performatively stable models satisfy

$$\theta_* \in \arg\min_\theta \mathbb{E}_{y \sim D(g(\theta_*))}\left[\ell(g(\theta), y)\right]. \tag{43}$$

The prediction dependent shift in targets is represented by the distribution $D(\cdot)$ being a function of the model predictions $g(\theta)$. Note that we have used the formulation of performative prediction with respect to model predictions (Mofakhami et al., 2023) instead of parameters.

The problem of minimizng projected Bellman error is a special case of (43) with $\ell(z, y) = ||z - y||_\Xi^2$ and $D(z)$ is the distribution with full weight on $T_\pi(z)$.

The problem of finding a performatively stable point can more generally be expressed as the following decision-dependent optimization problem:

$$\theta_* \in \arg\min_\theta f_{g(\theta_*)}(g(\theta)). \tag{44}$$

Or equivalently a constrained optimization problem with constraint $\mathcal{Z} = \mathrm{cl}\{g(\theta) : \theta \in \mathbb{R}^d\}$:

$$\theta_* \in \arg\min_\theta f_{g(\theta_*)}(g(\theta)) \Leftrightarrow z_* \in \arg\min_{z \in \mathcal{Z}} f_{z_*}(z).$$

If $Z$ is closed convex and $f_{z'}(z)$ is convex and differentiable with respect to $z$, then the above constrainted formulation is equivalent to the fixed point problem:

$$z_* = \Pi \circ T(z_*) = \Pi(z_* - \eta\nabla f_{z_*}(z_*)), \tag{45}$$

where we denote the gradient of $f_{z'}(z)$ with respect to $z$ as simply $\nabla f_{z'}(z)$. As mentioned in Section D, Bertsekas (2009) showed that solving (45) is equivalent to solving a strongly monotone

and smooth VI if either $T$ or $\Pi \circ T$ is a contraction. Therefore, under these contraction conditions (45) is a hidden strongly monotone and smooth VI problem of the form (1)! Thankfully, (Drusvyatskiy & Xiao, 2023; Cheng et al., 2020) showed that $\Pi \circ T$ is a contraction if $f_{z'}(z)$ is $\mu$ strongly convex with respect to $z$ and $\nabla f_{z'}(z)$ is $\beta$-Lipschitz with respect to $z'$ and if $\beta/\mu < 1$. Therefore, under standard performative prediction assumptions such as those found in Drusvyatskiy & Xiao (2023), the performatively stable point (43) is equivalent to solving a hidden strongly monotone and smooth VI problem where our surrogate loss approach (Algorithm (1)) applies and converges to the stable point under Theorems (3.2,3.6,A.5).

