# OpenReview forum: "Solving hidden monotone variational inequalities with surrogate losses"
_ICLR.cc/2025/Conference — ICLR 2025 Poster_

### Official Review · Reviewer_HyYK · 2024-10-29

**Soundness:** 3
**Presentation:** 2
**Contribution:** 3
**Rating:** 6
**Confidence:** 3

**Summary:**

This paper studies the problem of solving variational inequalities with deep learning methods with a hidden monotone structure. The authors present an iterative optimization algorithm with two nested loops, where In the outer loop, a surrogate square loss is constructed and partially minimized in the inner loop (until a sufficient decrease condition called $\alpha$-descent is satisfied) using an optimiser such as a quasi-newton method or ADAM. When $\alpha$ is sufficiently smaller than $1$, The authors prove linear (in the outer iterations) convergence guarantees in deterministic and stochastic settings, where in latter, the algorithm converges to a neighbourhood of the solution. They also prove that, when considering general variational inequalities, $\alpha < 1$ is not sufficient to guarantee convergence. Further, they show how several methods can be seen as special cases of their algorithm. They also present experimental results on min-max optimization and reinforcement learning.

**Strengths:**

1. The theoretical analysis is quite thorough, considering  deterministic and stochastic cases and also addressing an issue with a previous analysis. The authors also explain how previous methods fit in their framework and discuss assumptions and potential limitations.
2. Promising experiments showing that the method can achieve faster convergence with more than one inner step iteration also in practical settings such as reinforcement learning with MLP value functions.

**Weaknesses:**

1. Certain parts lack clarity. Condition 2 in Theorem 3.2 seems unnecessary and needs refinement. (See question and comments).
2. The paper lacks larger scale experiments. For example Deep RL experiments (with bigger underlying neural networks) could be included to demonstrate the claimed scalability of the method.

**Questions:**

Questions:
1. You claim that Sakos et al. (2024) assumes $\mathcal{Z}$ bounded implicitly in their main lemma. Can you clarify which lemma, where it is assumed and which results are affected? I could not easily find it and I think this is an important point since it uncovers a fallacy in a previous analysis.

Comments:
- The second condition on $\alpha$ in theorem 3.2 seems to be always satisfied by setting $p=1$ and $C‎ = \alpha/\eta$. From the proof it appears that there is an hidden dependency between $\eta$, $C$, $p$, and $\alpha$. The statement should either include such dependency or at least clarify this aspect.
- The use of the term “gradient step” contrasts with the Variational inequality formulation where in general $F$ is not a gradient. A possible alternative could be "proximal step".
- The related work paragraph in the introduction should probably be a subsection or its name should be removed or changed.
- The PL condition is not properly defined in the main text. It could be defined close to Line 345 or at least in the appendix.

References:

Sakos, Iosif, et al. "Exploiting hidden structures in non-convex games for convergence to Nash equilibrium." Advances in Neural Information Processing Systems 36 (2024).

---

> ### Author Response · Authors · 2024-11-19
> **Author Response**
>
> Thank you for you comments and questions, they will improve the paper. Below we have addressed all your raised concerns and look forward to your response.
>
> > Certain parts lack clarity. Condition 2 in Theorem 3.2 seems unnecessary and needs refinement. (See question and comments).
>
> We address condition 2 in more detail below.
>
> > The paper lacks larger scale experiments. For example Deep RL experiments (with bigger underlying neural networks) could be included to demonstrate the claimed scalability of the method.
>
> For our Mujoco RL experiments we selected a standard network architecture (2 hidden layer MLP) [3,4,5]. Our intention was not to select a small model but to demonstrate our approach in a standard Mujoco setup. However, we have included an experiment with 16 layers in stead of 2 to demonstrate the scalability of our approach, see Figure 10 in the appendix. Since in RL the bottleneck is mostly due to environment interaction we see comparable runtimes to the 2 layer experiments. Additionally the performance is similar, as expected due two-layers being sufficient.
>
> We would also like to emphasize that the scalability of our approach highly depends on the inner-loop optimizer. Since in practice GD with a small number of steps is sufficient (as demonstrated in our experiments and previous works), the surrogate loss approach is on the same order of complexity as SGD without a surrogate.
>
>
> > You claim that Sakos et al. (2024) assumes bounded implicitly in their main lemma. Can you clarify which lemma, where it is assumed and which results are affected?
>
> The set of predictions $\mathcal{Z}$ or as refered to by Sakos et al. the set of latent variables $\mathcal{X}$ is assumed to be bounded in Lemma 4, their "template inequality". More precisely, in the proof of Lemma 4 the existence of a constant $D = diam(\mathcal{X})$ (i.e. $\mathcal{X}$ is a set with bounded diameter) is used in equation B.20 in Appendix B. Lemma 4 is then used in Theorems 1,2,3,4.
>
> > The second condition on in theorem 3.2 seems to be always satisfied by setting $p=1$ and $C=\alpha/\eta$. From the proof it appears that there is an hidden dependency between $\eta$, $C$, $p$, and $\alpha$. The statement should either include such dependency or at least clarify this aspect.
>
> Thank you for this comment, we will clarify more precisely. What we meant exactly by this condition is that $\alpha$ can be made smaller by making $\eta$ smaller. That is
> $\exists C, p > 0$ such that $\forall \eta > 0$ it holds $\alpha < C\eta^p$.
>
> While the first condition is independent of $\eta$ the second one is more subtle. The example given above would not satisfy the condition since we require the inequality to hold for all $\eta$ (or more generally for all $\eta$ small enough).
> The intuition behind this condition is that the surrogate loss $\ell_t(\theta) = \frac{1}{2}\||g(\theta) - (g(\theta_t)-\eta F(g(\theta_t)))\||^2$ becomes easier as $\eta$ gets smaller allowing the current predictions $z_t=g(\theta_t)$ to start close to the exact PGD step $z_t^\ast$. We suspect some quasi-Newton methods may be able to take advantage of this structure and thought it to be useful to present this alternative condition on $\alpha$ as one way to quantify this.
>
> > The use of the term “gradient step” contrasts with the Variational inequality formulation where in general is not a gradient. A possible alternative could be "proximal step".
>
> We agree that this terminology can sometimes be confusing however it is common practice to refer to the update rule
> $$ z_{t+1} = z_t - \eta F(z_t)$$
> as the "gradient method" in the VI context.
> One example is the infamous extragradient (EG) method by Korpelevich. EG is mostly used in the VI context where there may not be a "gradient." In fact, in Korpelevich's seminal paper he refers to the update above as the "gradient method" [1].
> We disagree that the proximal method would be an appropriate description as it is usually meant to represent the following update rule:
> $$z_{t+1}= z_t -\eta F(z_{t+1}).$$
> See for example [2].
>
> > The related work paragraph in the introduction should probably be a subsection or its name should be removed or changed.
>
> We agree that this paragraph should be renamed, it would be more accurate and informative to specifically mention that this paragraph corresponds to **surrogate losses in scalar minimization**.
>
>
> > The PL condition is not properly defined in the main text. It could be defined close to Line 345 or at least in the appendix.
>
> Thanks for this comment we will add the PL definition in the appendix.

---

> > ### Author Response · Authors · 2024-11-19
> > **Author Response Continued**
> >
> > References:
> >
> > [1] Korpelevich, G. M.: The extragradient method for finding saddle points and other problems, Matecon 12 (1976), 747–756.
> >
> > [2] Mikhail V Solodov and Benar F Svaiter. A hybrid approximate extragradient–proximal point algorithm using the enlargement of a maximal monotone operator. Set-Valued Analysis, 7(4):323–345, 1999.
> >
> > [3] Haarnoja T, Zhou A, Abbeel P, Levine S. Soft actor-critic: Off-policy maximum entropy deep reinforcement learning with a stochastic actor. InInternational conference on machine learning 2018 Jul 3 (pp. 1861-1870). PMLR.
> >
> > [4] Achiam J, Knight E, Abbeel P. Towards characterizing divergence in deep q-learning. arXiv preprint arXiv:1903.08894. 2019 Mar 21.
> >
> > [5] Achiam J. Benchmarking reinforcement learning algorithms. Spinning Up in Deep RL. OpenAI; Available from: https://spinningup.openai.com/en/latest/spinningup/bench.html. Accessed 2024 Nov 18.

---

> ### Comment · Reviewer_HyYK · 2024-11-23
>
> Thank you for the response. I am still dubious about some parts of the submission.
>
> >What we meant exactly by this condition is that $\alpha$ can be made smaller by making $\eta$ smaller. That is $\exists C, p > 0$ such that $\forall \eta > 0$ it holds $\alpha < C\eta^p$.
>
> Theorem 3.2 mentions at the start "there exist a stepsize $\eta$" so the condition holding for all $\eta$ is misleading.
> Moreover, the proposed condition seems never satisfied: for every $\alpha, C, p > 0$ there exist $\eta > 0$ such that $\alpha > C \eta^p$, for example if we take $\eta = (\alpha/(2C))^{1/p}$, then $C \eta^p = \alpha/2 < \alpha$.
>
> From looking at the proof it seems that $\eta$ and $\alpha$ are linked and should satisfy some conditions depending on the problem, why do you not directly state those conditions in the theorem statement? Alternatively, I suggest you to remove the second condition, also because Line 830-836 in the proofs are not clear and seems that a step is missing: verifying that $\eta$ is not so small to violate the inequality $\alpha < C\eta^p$ assumed at the start of the derivation.
>
> >The set of predictions $\mathcal{Z}$  or as refered to by Sakos et al. the set of latent variables $\mathcal{X}$  is assumed to be bounded in Lemma 4, their "template inequality". More precisely, in the proof of Lemma 4 the existence of a constant...
>
> Could you include this statement in the manuscript? I think this is subtle, while should be clear to readers.
>
> >For our Mujoco RL experiments we selected a standard network architecture (2 hidden layer MLP) [3,4,5]. Our intention was not to select a small model but to demonstrate our approach in a standard Mujoco setup. However, we have included an experiment with 16 layers in stead of 2 to demonstrate the scalability of our approach, see Figure 10 in the appendix. Since in RL the bottleneck is mostly due to environment interaction we see comparable runtimes to the 2 layer experiments. Additionally the performance is similar, as expected due two-layers being sufficient.
>
> Thank you for providing additional experiments. However, as you also mention, this setup looks very artificial and 2 layers are already enough. I would have preferred more challenging problems really taking advantage of the larger networks, sorry for not having specified this in the review. This part remans still a minor weakness of the work.

---

> > ### Author Response · Authors · 2024-11-28
> > **Author Response**
> >
> > We would like to thank you again for your detailed comments and feedback. We believe our recent edits will improve the clarity of the paper and our contribution overall.
> >
> > ## Conditions in Theorem 3.2
> >
> > > Theorem 3.2 mentions at the start "there exist a stepsize $\eta$" so the condition holding for all $\eta$ is misleading. Moreover, the proposed condition seems never satisfied: for every $\alpha, C, p > 0$ there exist $\eta >0 $ such that $\alpha > C\eta^p $ , for example if we take $\eta = (\alpha/(2C))^{1/p}$, then $C\eta^p = \alpha/2 <\alpha$.
> > >
> > > From looking at the proof it seems that $\eta$ and $\alpha$ are linked are linked and should satisfy some conditions depending on the problem, why do you not directly state those conditions in the theorem statement?
> >
> > You are absolutely correct when you wrote that $\eta$ and $\alpha$ were linked by the condition
> > $$
> > 1-2\eta(\mu-\alpha L) + (1+\alpha^2)\eta^2 L^2 <1
> > $$ we will state that in the theorem instead. The goal of condition 1) (or 2)) was to give a some sufficent condition for which the equation above was holding. We agree with the reviewer suggestions regarding how to improve the presentation of Theorem 3.2. which was updated as follows:
> >
> > **Theorem 3.2**:   Let Assumption 3.1 hold and let $\(z_t =g(\theta_t)\)$ be  the iterates produced by Algorithm 1. If $\alpha$ and $\eta$ are picked  such that,
> > $$
> > \rho:= 1-2\eta(\mu-\alpha L) + (1+\alpha^2)\eta^2 L^2 <1
> > $$ then, $z_t$ converge linearly to the solution $z_\ast$ at a rate $O(\rho^t)$. In particular, if $\alpha = \frac{\mu}{2L}$ and $\eta = \frac{2\mu}{5L^2}$ we have $\rho \leq 1 - \frac{\mu^2}{5L^2}$.
> >
> >
> > ## Bounded assumption of Sakos et al
> >
> > > Could you include this statement in the manuscript? I think this is subtle, while should be clear to readers.
> >
> > We have added the mentioned statement as a footnote on page 7.
> >
> > ## Deep RL Experiments
> >
> > > Thank you for providing additional experiments. However, as you also mention, this setup looks very artificial and 2 layers are already enough. I would have preferred more challenging problems really taking advantage of the larger networks, sorry for not having specified this in the review. This part remans still a minor weakness of the work.
> >
> > Thank you for this clarification. We would like to emphasize that our main contribution lies in providing a theoretical framework for analyzing convergence in variational inequality problems with hidden structure. Based on this framework, we proposed a novel method for value prediction tasks in deep RL.
> >
> > We do not believe the current setup to be artificial, Mujoco is a well-established benchmark for deep RL tasks, used in RL papers published at top-tier conferences including ICLR [1,2,3]. Additionally, our results on these tasks demonstrate significant performance improvements, particularly in terms of data efficiency and runtime, which we believe are valuable to the RL community.
> >
> > We appreciate your concern on scalability, however. We believe with our 16 layer experiment we have shown our approach to be scalable to much larger networks (like we have emphasized in the paper). We acknowledge that it will be interesting to extend our methods to more challenging tasks that require larger networks (e.g. Estimation of the Q function of a Chess engine [4]) but we believe it not reasonnably achieveable within the timeframe of the discussion period and is outside of the scope of this paper. However, we plan to work on such ideas and applications as a follow-up.
> >
> >
> > [1] Manan Tomar, Lior Shani, Yonathan Efroni, and Mohammad Ghavamzadeh. Mirror descent policy optimization. In International Conference on Learning Representations, 2022.
> >
> > [2] Vaswani, S., Bachem, O., Totaro, S., Müller, R., Garg, S., Geist, M., ... & Le Roux, N. (2022, May). A general class of surrogate functions for stable and efficient reinforcement learning. In International Conference on Artificial Intelligence and Statistics (pp. 8619-8649). PMLR.
> >
> > [3] Fujimoto, S., Hoof, H., & Meger, D. (2018, July). Addressing function approximation error in actor-critic methods. In International conference on machine learning (pp. 1587-1596). PMLR.
> >
> > [4] Farebrother, Jesse, et al. "Stop regressing: Training value functions via classification for scalable deep rl." ICML 2024.

---

### Official Review · Reviewer_L5s9 · 2024-10-31

**Soundness:** 2
**Presentation:** 2
**Contribution:** 4
**Rating:** 6
**Confidence:** 3

**Summary:**

The paper proposes an algorithm using some type of surrogate approximation to solve variational inequality (VI) problems. Those problems can be seen as finding first-order stationary points of a constrained optimization problem, but probably the setting in this paper is more general than that since the vector-valued function F is not necessarily a gradient (e.g. max-min game). The main idea is that "composite" (between the model and the loss) optimization problems in machine learning normally exhibit some structure, e.g., the loss w.r.t. to the model's output is convex (but the whole optimization function is not convex w.r.t. the model's parameters), one can push the "difficulty part" relating to model into the constraint to make the objective function convex. The authors then design a sequence of surrogate functions to minimize this reformulation problem and show convergence under a condition called "alpha-descent". To minimize the surrogate functions, they employ classical methods like Gaussian-Newton or Levenbergh-Marquardt. Numerical experiments are performed for some toy min-max games and in the reinforcement learning context.

**Strengths:**

The considered problem is important in the classical optimization context (i.e., constrained optimization, complementarity) and mordern ML where the loss is structured. The problem is also more general than minimizing a scalar loss usually showing up in supervised learning. The experiments show that the proposed method work fine in practice.

**Weaknesses:**

The paper is challenging to follow, particularly in its transition from problem (1) to the construction of the surrogate model, where additional discussion would be beneficial. The assumptions also seem overly restrictive. For instance, while assuming convexity of the loss with respect to the model's output is reasonable for most loss functions, the assumption that the constrained domain is convex feels unnecessarily limiting, even though the authors provide a few narrow examples. Furthermore, the alpha-descent condition (5) requires closer examination, as it appears to be a stringent requirement. Specifically, it requires a single constant alpha that holds uniformly across all t.

**Questions:**

(1) In the basic case of supervised learning with a scalar loss, can we expect the proposed method perform better than off-the-shelf optimizers that work directly in the parameter space, i.e., Adam?

(2) The condition in the while loop of Algorithm 1 can not be verified. How could we let alpha be the user-defined parameter?

---

> ### Author Response · Authors · 2024-11-19
> **Author Response**
>
> Thank you for your review. Below we address your comments and concerns in detail. We believe surrogate losses have played an important role in modern ML such as in policy gradient methods in reinforcement learning and we are excited to bring this approach to the VI setting. Despite the limitations that we have outlined in the paper we believe our results to be an important contribution to:
> 1. understand why surrogate loss methods work with a small number of gradient steps
> 2. understand why we should or shouldn't expect them to work in the VI setting if at all
>
> In doing so we have shown:
> 1. An optimizer agnostic approach to reducing VI problems to scalar minimization problems with convergence guarantees. Our approach does not assume what optimizer is used and is compatible with any deep learning optimizer. We discuss methods like Gauss-Newton to show how existing methods are a special case of our approach.
> 2. Provide a novel surrogate loss approach to learning value functions in RL and demonstrated a **significant improvement over data efficiency** when compared to TD(0), a standard approach to value learning
> 3. We have provided a non-trivial example to show that surrogate losses in VI problems are strictly more difficult than scalar minimization
>
>
> > The paper is challenging to follow, particularly in its transition from problem (1) to the construction of the surrogate model, where additional discussion would be beneficial.
>
> We devote most of Section 2 "Surrogate Loss Background" to discuss how problem (1) is related to different examples like supervised learning and min-max problems. In this section, we also discuss the intuitions behind the surrogate approach (i.e. approximating a gradient step in the prediction space) and why this is beneficial when there is hidden structure. We also connect all of the different components in problem (1) to these applications (e.g. $g$, $\mathcal{Z}$, and $F$).
>
>
> > The assumptions also seem overly restrictive. For instance, while assuming convexity of the loss with respect to the model's output is reasonable for most loss functions, the assumption that the constrained domain is convex feels unnecessarily limiting, even though the authors provide a few narrow examples.
>
> Thank you for this comment, indeed this is a restriction of our analysis as we have outlined in Section 2. However, we would like to highlight a few points regarding this assumption and believe that our results even with this assumption are an important step to extending the surrogate approach to VI problems.
>
> + We linked to two interesting extreme cases when $\mathcal{Z}$ is convex: (1) when a linear model is used, and (2) when the model is large enough to interpolate any dataset. In fact (2) has been shown to not be so extreme where large capacity neural networks are able to interpolate random noise [2].
> + We provide the first extension of surrogate methods to VI problems with convergence to the global solution of the VI. In contrast, previous works that achieved similar results are **only in the scalar minimization case** where **they also assume the set of predictions is convex** [1].
> + This assumption in fact **strengthens** our negative result since even when $\mathcal{Z}$ is convex, divergence is possible for $\alpha<1$
>     + This is surprising since $\alpha <1$ is enough for convergence in scalar minimization even without convexity of the loss or the set of predictions! This demonstrates how much more difficult it is to theoretically analyze a surrogate approach in the VI case.
>
>
>
>
> > Furthermore, the alpha-descent condition (5) requires closer examination, as it appears to be a stringent requirement. Specifically, it requires a single constant alpha that holds uniformly across all t
>
> The alpha condition indeed needs to hold for all $t$, however, we believe it is actually less stringent than existing approaches that require the error $\ell_t(\theta_{t+1})-\ell_t^\ast < \epsilon$ to be bounded by a small constant $\epsilon$ across all $t$ (e.g. [1]).
> Our $\alpha$ descent condition provides an improvement on multiple fronts:
>
> + The $\alpha$-descent condition is a more accurate model than just bounding the error when a moderate amount of optimization is used at each $t$ (e.g. a small number of gradient steps). In Section 4 we highlight some cases.
> + We achieve convergence to the solution $z_\ast$ without assuming the errors $\||z_{t+1}-z_t^\ast\||$ to be summable  or going to zero a priori, which is common even in the scalar minimization case (see the related work paragraph in Section 2 lines 146-155 for discussion).

---

> > ### Author Response · Authors · 2024-11-19
> > **Author Response Continued**
> >
> > > (1) In the basic case of supervised learning with a scalar loss, can we expect the proposed method perform better than off-the-shelf optimizers that work directly in the parameter space, i.e., Adam?
> >
> > In our paper we have focused on the VI setting and shown significant data efficiency improvements by using a surrogate loss approach. **We would also like to emphasize that our approach is optimizer agnostic. Our surrogate loss approach actually allows one to use Adam in the inner loop if they so choose.** If we take one gradient step instead of multiple steps in the inner loop then our method is exactly gradient descent where the stepsize is $\eta\cdot\eta_{alg}$ where $\eta$ is the stepsize in the surrogate loss and $\eta_{alg}$ is the stepsize used by the gradient step in the inner loop. Therefore our approach includes Adam as a special case. [1] studied the use of surrogate losses with a fixed stepsize inner loop in the supervised learning case and showed comparable results to Adam. It is unclear if they tried Adam in the inner loop of their surrogate method. In our RL experiments we used AdamW in our innerloop and found it to outperform AdamW with one step (i.e. TD(0)).
> >
> > > (2) The condition in the while loop of Algorithm 1 can not be verified. How could we let alpha be the user-defined parameter?
> >
> > In the paper we outline cases and several approaches to deal with this:
> > *  In Section 4 we discuss some cases where a finite number of gradient steps guarantees any $\alpha$ therefore you can just tune the number of gradient steps
> > * In a highly over parameterized regime, $\ell_t^\ast$ is 0 and therefore $\alpha$ can be checked directly. We can then optimize the inner loop until it is satisfied.
> > * In practice a finite number of gradient steps works well practice across a wide range of tasks
> >     * Min max optimization (our paper)
> >     * Minimizing projected Bellman error (our paper)
> >     * Supervised learning [1]
> >     * Policy gradient RL [3]
> >
> > In comparison to all the existing works, despite ours being in the more difficult VI setting, our descent condition is the closest to understanding why a small number of gradient steps can converge to a good solution.
> >
> > References:
> >
> > [1]  Lavington, J.W., Vaswani, S., Babanezhad Harikandeh, R., Schmidt, M. &amp; Le Roux, N.. (2023). Target-based Surrogates for Stochastic Optimization. <i>Proceedings of the 40th International Conference on Machine Learning</i>, in <i>Proceedings of Machine Learning Research</i> 202:18614-18651 Available from https://proceedings.mlr.press/v202/lavington23a.html.
> >
> > [2] Chiyuan Zhang, Samy Bengio, Moritz Hardt, Benjamin Recht, and Oriol Vinyals. Understanding deep learning requires rethinking generalization. In International Conference on Learning Representations, 2017. URL https://openreview.net/forum?id=Sy8gdB9xx.
> >
> > [3] Sharan Vaswani, Olivier Bachem, Simone Totaro, Robert Müller, Shivam Garg, Matthieu Geist, Marlos C. Machado, Pablo Samuel Castro, Nicolas Le Roux Proceedings of The 25th International Conference on Artificial Intelligence and Statistics, PMLR 151:8619-8649, 2022.

---

> > > ### Comment · Reviewer_L5s9 · 2024-11-22
> > >
> > > Thank you for your detailed response. My main concern which is the restriction of the assumptions still stays. However, regarding the reply and taking into account that other related works also have more or less the same restricted assumptions, I believe this direction of research is still in progress and -- therefore -- am willing to increase my overall score to 6 and the contribution score to 4.

---

### Official Review · Reviewer_eBkP · 2024-11-03

**Soundness:** 4
**Presentation:** 3
**Contribution:** 4
**Rating:** 8
**Confidence:** 4

**Summary:**

In this paper, the authors introduce a surrogate-loss-based framework for optimization of variational inequality (VI) systems, such as min-max.

This paper is the first to extend the surrogate methodology to (strongly-monotone) VIs, where the inner loop calls a scalar optimization routine black-box. Importantly, they demonstrate with an elementary adversarial example that surrogate methods in these VIs are qualitatively more complex than surrogate methods in traditional scalar optimization (the inner loop progress needs to be strong enough to counteract the effects of $F$ to ensure outer convergence, as opposed to just $<1$ in scalar case). They show that (under sufficient inner loop optimization assumptions, quantified in the form of the $\alpha$-descent condition) the overall VI optimization succeeds in deterministic and stochastic regimes, with rates matching what is to be expected of strongly-convex optimization (ie geometric convergence). Lastly, they observe that an existing VI optimization method (PHGD) can be seen as a particular choice of inner loop optimization routine, and they investigate the benefits and consequences of alternative choices of inner loop algorithm and number of steps.

Experimentally, the authors test the surrogate method for VI optimization in previously-investigated small-scale games, as well as on value prediction in fairly substantial RL settings. They observe interesting consequences of certain choices of inner loop algorithm and number of steps, and they demonstrate the value of the surrogate framework. In the RL experiments they see a significant improvement in environment sample complexity due to multiple inner loop iterations -- this matches some related work, but importantly does not require complex learned surrogate reward signals!

To summarize, the paper introduces the surrogate framework (outer loop over outputs and inner loop over model parameters) to a class of VI problems, demonstrates the nontrivial result that scalar optimization can (under some assumptions) be used black-box as a subroutine, and empirically investigate the associated phenomena on tasks of increasing complexity. Very cool!

**Strengths:**

This paper has a strong originality in that it appears to be the first to extend this type of surrogate losses to VIs, and it does so in a way that provably takes advantage of hidden monotonicity/convexity structure. Importantly, this extension is nontrivial -- there is a simple and solid adversarial example the authors provide (Prop. 3.3) that shows a difficulty gap in comparison with surrogate losses in the scalar optimization case. I think there is the extra strength (though perhaps it isn't highlighted enough, see my "Weaknesses" section) that it appears the framework, broad proof techniques, and main takeaways seem rather robust to choice of inner optimization routine (i.e. robust to assumptions, rates of convergence, etc.). In my eyes, the authors have constructed a fairly general methodology to reduce VIs with hidden structure to scalar optimization in a way that directly leverages that hidden structure -- this is very cool, and in general such reduction results tend to be quite powerful (such as online-to-batch reduction, nonconvex-to-convex reduction such as Algorithm 3 in https://arxiv.org/pdf/1806.02958, etc).

The analysis is also quite clear, proceeding along similar lines to many optimization papers and doing a fantastic job of contextualizing results, definitions, and assumptions in prior works. In particular, Section 4 (before subsection 4.1) skillfully highlights the flexibility of choices of inner loop optimizers in an organized way, noting equivalence to prior methods (such as PHGD) where applicable. This is a good transition to the experiments, which compare different setups of the inner loop routine in various minimax and RL value-prediction environments. Overall, I think this presentation is clear, the assumptions are obvious, and the experimental apparatus seems right for the general framework (though it would have been cool to see a slightly larger-scale experiment, and I think GAN training is the perfect low-hanging fruit if the authors have the resources!). Lovely stuff :)

**Weaknesses:**

I think the main (and only, to be honest) weakness of this paper is a weakness of presentation -- in particular, I feel that (as outlined in the "Strengths" section of my review above), the main contribution of this paper is that it clarifies and formalizes a framework where black-box scalar non convex optimization guarantees can be bootstrapped up to hidden-structure VI guarantees. However, at many points I felt that the presentation did not highlight this strongly enough, and instead chose to focus on particular rates/modes of convergence and substantiating particular assumptions.

To be specific, I would argue that the $\alpha$-descent condition for inner loop progress is a bit of a red herring. As you mention, such convergence can only be shown in specific situations under the usual non convex optimization assumptions (PL condition, for example), which can often be difficult to justify. However, I feel that it's even unnecessary for you to justify it! It seems to me that, for example, Lemma A.2 and Prop A.3 would go through (perhaps with significantly more work) for weaker/more exotic types of inner loop guarantee -- the vibe of the proofs is moreso that (strongly-monotone) VIs allow you to push through the classic optimization proofs (the [Sakos '24] paper offers a similar takeaway in terms of bootstrapping convex VI guarantees up to hidden-convex VI guarantees, see their statements on p. 11). I bet there is a way to turn this into a strength of your paper: maybe something more like "we prove things under the $\alpha$-descent condition for clarity, but our meta-point is a more general reduction from VI optimization to scalar optimization via surrogate losses". I am not recommending you to do the analysis under all kinds of crazy inner loop guarantees, but instead to reweight the presentation a bit to highlight the robustness to inner loop method.

I will say that the $\alpha$-descent setting is a fantastic choice of inner loop guarantee to demonstrate the difficulty gap between scalar vs VI surrogate methods; it makes the presentation of the adversarial example very clear. However, I would have liked to see it used more as a presentation tool/particular instantiation of a more general phenomenon, whereas it often felt like you were keeping it around as a core part of your results. If the impact of this condition was qualitatively different in the VI setting than in scalar surrogate optimization then that would be one thing, but I am unsure of this (note: I am not too familiar with this style of surrogate losses via hidden convexity/monotonicity -- if I am wrong about this perhaps a toy example exemplifying the difference would be cool!).

To really hammer this point home (sorry, but I don't really see any other weaknesses to write about), I feel like over-indexing on this particular criterion forces you to get stuck in the muck of justifying assumptions such as spectrally bounding input-output Jacobians of neural networks -- to some this may be a losing battle (models on complex, high-dim data will be likely to disregard features dynamically and hopefully learn lower-rank representations), but one I don't think you need to be fighting! Certain hypothesis classes/optimizers/datasets/etc will have different choices of inner loop routine that make sense, and the beauty of what you've shown here is that surrogate methods in VIs appear flexible to these choices. The language of reductions feels much more natural for such a result: I give you a VI problem with hidden monotone structure, you reduce it to a sequence of scalar non convex problems, and the practitioner/domain expert figures out the right inner loop soup (choosing between first-order or second-order methods, bias-variance tradeoffs, etc) to make it work.

To sum up, it is my opinion that if you are going to use non convex optimization as a black-box for the inner loop, treat it like a black-box (not just in terms of whether to use ADAM as the optimizer, but even in a broader sense). Aside from this (and some questions that I put in the "Questions" section), I have nothing else to say but awesome paper!

**Questions:**

1. I think it's very interesting that sometimes more inner loop iterations damage the performance as a whole! (I once did some empirics for boosting weak learners in RL, where I saw a similar thing that I attributed to overfitting...). Do you have any explanations/intuitions for why this could happen (ie is it a general phenomenon in surrogate learning, or an artifact of these small-scale game experiments)?

2. The results on improved sample complexity in the RL value prediction tasks are very compelling in my opinion! I think it fits neatly into an overarching philosophy that blind RL is wasteful (in terms of samples/environment interactions), and that some form of guidance really helps. There are whole fields (search goal-conditioned RL, contrastive RL, etc) that attempt to figure out how to learn the right flavor of guidance, and it seems to me that your form of (not-learned!) surrogate losses can be seen as a particularly simple form of this. From your work and the related literature (which you know 1000x better than I), do you suspect that there can be any theoretical insight/proofs on synthetic settings for how surrogate losses get a provable advantage in terms of number of diverse samples? Certainly one can make claims about decreasing the # of outer loop iterations, but I would be very interested if the extra regularity of your simple surrogate loss trajectory (ie the GD trajectory on $z_t$) can manifest as more efficient exploration?

3. Probably not a valuable question, but I have to ask: would it be possible for you to do GAN training? It would be convincing to deep learning practitioners and theorists alike!

3. This is more a question of personal interest regarding this style of surrogate losses (i.e. ones where you take a step in output space, use inner loop to make the model catch up, repeat) and perhaps not specific to VIs, but here goes: *is there any understanding of how this framework assists/counteracts the implicit biases of either (1) optimization methods in the inner loop or (2) architectural choices in the model $g$?* I ask because, particularly in deep learning theory, there is often the vibe that the natural dynamics induced by such design parameters actually can help (see this paper https://arxiv.org/pdf/1802.06509 for a cool result on GD acceleration caused by model depth, for example). I could imagine some settings where the rigidity of the outer loop dynamics on $z_t$ prevent these complicated phenomena (for example, in the linked paper I suspect surrogate losses could prevent the acceleration for adversarial choices of $F$). Conversely, I can certainly imagine settings where the structure of the outer loop drives the optimization more usefully, in a similar fashion to how surrogate rewards in RL help orient an agent in a sparse-reward environment (see Question 2). Is there any understanding of this tradeoff, and perhaps more importantly do you imagine any differences to this tradeoff in the VI setting?

---

> ### Author Response · Authors · 2024-11-19
> **Author Response**
>
> We are happy to hear that you found our paper interesting, well-written and our contribution both original and nontrivial. We also appreciate the detailed questions and potential directions for future work. We are hoping to set a foundation for which others can build better algorithms for difficult VI problems, and are excited to see many directions that we have not considered.
>
>
> > I think it's very interesting that sometimes more inner loop iterations damage the performance as a whole! (I once did some empirics for boosting weak learners in RL, where I saw a similar thing that I attributed to overfitting...). Do you have any explanations/intuitions for why this could happen (ie is it a general phenomenon in surrogate learning, or an artifact of these small-scale game experiments)?
>
> Thank you for this comment, reviewer LPnR also highlighted this observation. We also found it surprising and believe it to be important to investigate further as future work. As we mentioned to reviewer LPnR, we believe this behaviour to highly depend on the problem and might be an artifact of our small dimensional toy problems. Given our current black box treatment of the inner loop it is difficult to understand theorertically from our analysis why such a phenomenon would occur. As mentioned by reviewer LPnR, it is possible a different descent condition could explain this behaviour. On the other hand, it is also possible that the performance is explainable via the $\alpha$-descent condition but with respect to a different method that is faster than the projected gradient method.
>
> >do you suspect that there can be any theoretical insight/proofs on synthetic settings for how surrogate losses get a provable advantage in terms of number of diverse samples? Certainly one can make claims about decreasing the # of outer loop iterations, but I would be very interested if the extra regularity of your simple surrogate loss trajectory (ie the GD trajectory on $z_t$) can manifest as more efficient exploration?
>
> In our experiments we focused on value prediction i.e. learning a value function. Where the behavioural policy used to collect the trajectories is fixed. In this case the methods used would not change the trajectories. There is some mention in [2] on how surrogate losses can affect exploration in policy gradient methods, however, in general this seems to be an open question.
>
> > would it be possible for you to do GAN training?
>
> We think GANs are an interesting direction for future work. However, from our understanding, most GAN losses do not admit a tractable hidden monotone structure (i.e. a hidden convex-concave game). Let us provide more details. Given a disciminator $D_w$ and a generator $G_\theta$, the standard GAN minimax loss is:
> $$\min_\theta \max_w E_{x\sim p_{data}(x)}\log D_w(x) + E_{z\sim p_z(z)}\log (1- D_w(G_\theta(x))$$
> which is concave with respect to the discriminator $D$ but **not convex** with respect to the generator function $G$ but can be seen as a convex-concave minimax loss with respect the the generated distribution $p_G$:
> $$ \min_w \max_\theta E_{x\sim p_{data}(x)}\log D_w(x) +E_{x'\sim p_{G_\theta}(x')}\log (1- D_w(x')
> $$
> However, we cannot compute $\nabla_\theta E_{x'\sim p_{G_\theta}(x')}\log (1- D_w(x'))$ easily with the standard reinforce trick because it require to compute $\nabla_\theta \log p_{G_\theta}(x')$ which is not tractable for most GAN as they do not have explicit density function (which is considered to be one of their important features: https://arxiv.org/pdf/1701.00160). Thus, developping GANs that allow for a tractable surogate-based optimization leveraging their hidden convex-concave formulation is outside of the scope of this paper. Instead we decided to focus on RL applications.
>
>
> Meanwhile, minimizing projected Bellman error was shown by Bersekas [1] to be a hidden smooth and strongly monotone problem, the exact setting we study in our paper. Additionally, given the constraints of a conference paper, we believe it was important to share Bertsekas' VI perspective of projected Bellman error to the deept RL community and show:
> 1. that it naturally allows for a surrogate approach similar to standard policy gradient methods like PPO [3]
> 2. draw connections to our perspective by showing it as a special case of the PHGD method and more generally using Gauss-Newton as an inner-loop optimizer. Note that the surrogate interpretations of this method as given by equation (13) is novel.

---

> > ### Author Response · Authors · 2024-11-19
> > **Author Response Continued**
> >
> > > is there any understanding of how this framework assists/counteracts the implicit biases of either (1) optimization methods in the inner loop or (2) architectural choices in the model $g$?
> >
> > Thank you for this question, we think it would be exciting future work to better understand the interplay between model choices and inner-loop optimizers. In our paper we outline two cases  cases where the choice of $g$ gives us extra insight. In the linear case, we know that one-step of Gauss-Newton in the inner-loop (i.e PHGD) is sufficient since $\alpha = 0$ (as noted by [1]). If the model $g$ is sufficiently large enough where $\mathcal{Z}=\mathbb{R}^n$ then $\ell_t^\ast = 0$ and $\alpha$ can be easily monitored. It would be interesting to investigate as future work how certain architecture/ inner loop optimizer pairs interact for specific cases such as in the paper you shared.
> >
> >
> > References:
> >
> > [1] Dimitri P Bertsekas. Projected equations, variational inequalities, and temporal difference methods. Lab. for Information and Decision Systems Report LIDS-P-2808, MIT, 2009.
> >
> > [2] Sharan Vaswani, Olivier Bachem, Simone Totaro, Robert Müller, Shivam Garg, Matthieu Geist, Marlos C. Machado, Pablo Samuel Castro, Nicolas Le Roux Proceedings of The 25th International Conference on Artificial Intelligence and Statistics, PMLR 151:8619-8649, 2022.
> >
> > [3] John Schulman, Filip Wolski, Prafulla Dhariwal, Alec Radford, and Oleg Klimov. Proximal policy optimization algorithms. arXiv preprint arXiv:1707.06347, 2017.

---

### Official Review · Reviewer_LPnR · 2024-11-03

**Soundness:** 4
**Presentation:** 3
**Contribution:** 3
**Rating:** 8
**Confidence:** 3

**Summary:**

This paper introduces a new method for optimizing variational inequalities with hidden structure by optimizing a series of surrogate losses, thereby extending previous methods for optimizing scalar loss functions. The authors provide a new $\alpha$-descent condition on the sequence of inner surrogate optimization problems which is used to derive linear convergence rates for the outer optimziation in the deterministic and unconstrained stochastic setting. Specific choices of optimizer for the inner surrogate loss are shown to generalize previous works. Additionally, the authors provide conditions under which linear convergence is achieved.

Experimentally, the method is tested on optimizing min-max games and projected Bellman error. In the min-max setting different variants of the method are compared, showing that the choice of the inner optimizer matters by improving on the special cases treated in previous work. In the RL setting, the surrogate loss perspective is connected to computationally expensive preconditioning methods which is shown to be approximated in the linear case via the presented iterative scheme. In the non-linear case the policy evaluation problem is tackled for two mujoco environments, where different versions of the method are shown to improve over the special case of TD(0) in terms of wall-clock time and sample efficiency.

**Strengths:**

- The extension of iterative surrogate optimization from the scalar to the variational inequality case is a significant contribution.
- Paper offers a rigorous theoretical analysis.
- Strong performance of the method compared to the TD0 baseline.
- Well-written and relatively easy to follow.

**Weaknesses:**

- The empirical finding of better optimization of the inner problem not leading to better optimization of the outer loop is very interesting but unfortunately not examined in more detail. Both a more in-depth experimental investigation and a theoretical justification for this effect could strongly improve the paper, see also the questions below.

**Questions:**

- It is very interesting that better convergence of the inner loop does not necessarily translate to better convergence of the outer loop, i.e. more iterations are not necessarily useful (e.g. LM, Sur-GD, GN, Gig 1 & 2). Is there a theoretical justification? How does that tie in with the $\alpha$-descent rule? If not a low loss value, what makes a "good" solution to the inner problem that improves convergence of the outer loop? Has this effect also been observed in the scalar minimization case? If yes, how does it compare?
- The authors write: "In general $\ell_t^*$ may not be zero and so this condition cannot be verified directly, however, this condition can often be met via first-order methods for a fixed number of steps or can be approximated with $\ell_t^*=0$." (line 171) I am wondering how practical $\alpha$-descent condition is, is it possible to verify this condition in the presented experiments in section 5?

Style:
- Fig 1: It is hard to tell the methods apart, maybe use different line styles.

---

> ### Author Response · Authors · 2024-11-19
> **Author Response**
>
> We are pleased to hear that you found our paper well-written and our contributions significant. We are hoping that our work opens the door to new methods in solving difficult modern VI problems and at the same time provides new insights to existing approaches.
>
> Thank you for your comments and feedback. We address your comments and questions below in detail.
>
> > The empirical finding of better optimization of the inner problem not leading to better optimization of the outer loop is very interesting but unfortunately not examined in more detail. Both a more in-depth experimental investigation and a theoretical justification for this effect could strongly improve the paper, see also the questions below.
>
> Thank you for this comment. We indeed found this surprising and interesting. However, we did not see such behaviour in our larger scale experiments where more inner loop iterations resulted in better performance (with respect to the outer loop). Similarly, for supervised learning [1] also observed that more inner steps improved performance. Therefore, this behaviour observed in our experiments may be due to the small dimensionality of the toy problems in Section 5.
>
> Since we believe this behaviour is both instance and optimizer specific (i.e. depending on the problem and the optimizer), we believe it to be out of the scope of this paper. Our current focus and approach is optimizer agnostic -- with any optimizer  being used in the inner loop. However we agree that potential insight might be possible by considering a new descent like conditions (see below).
>
> > How does that tie in with the  $\alpha$-descent rule? If not a low loss value, what makes a "good" solution to the inner problem that improves convergence of the outer loop?
>
> Since our approach has been to consider the inner loop as a black-box and we do not use any knowledge of the algorithm inside to derive our convergence guarantees, it is difficult to pinpoint how exactly this phenomenon corresponds to   $\alpha$-descent. It is possible that in these examples the methods are related to $\alpha$-descent but with respect to a different method that is faster than the projected gradient method. Alternatively, as you mention, $\alpha$-descent may not be the best way to measure a good solution of the inner loop. We believe however that the $\alpha$-descent condition is a natural choice and an important starting point since it corresponds to the "gap" of the scalar loss in the inner loop that is often studied in convex or non-convex optimization. In addition, we believe that the $\alpha$-descent condition is an important step in analyzing surrogate methods since existing approaches require bounding the errors or gap by a small constant (e.g. [1]) instead of using a relative descent condition like ours. In comparison we believe our approach is a more accurate model of what is used in practice e.g. a moderate amount of gradient steps without forcing a small loss value in each inner loop (see more details below).
>
> > Has this effect also been observed in the scalar minimization case? If yes, how does it compare?
>
> [1] did not observe this behaviour in supervised learning. [2] looked at surrogate losses for policy gradient methods in RL and showed that more gradient steps in the inner-loop can hurt performance in some environments. However, we do not know if the surrogate loss value was smaller when more steps were used since this statistic was not reported. Furthermore, the surrogate loss is stochastic and is biased, an issue not present in our small scale deterministic setting. Therefore we don't know if (1) more steps is actually better minimizing the loss like in our case or (2) if this is an issue due to bias. For more details about this bias see [1] or our Section 5.2.2.
>
> > The authors write: "In general may not be zero and so this condition cannot be verified directly, however, this condition can often be met via first-order methods for a fixed number of steps or can be approximated with." (line 171) I am wondering how practical $\alpha$-descent condition is, is it possible to verify this condition in the presented experiments in section 5?
>
> In general without knowing $\ell_t^\ast$ it is not possible to verify the $\alpha$-descent condition directly. However, in some cases a fixed number of GD steps is sufficient to guarantee $\alpha$-descent. In Section 4 we provide some discussion on when a fixed number of inner loop gradient descent steps can be used to satisfy any $\alpha$-descent condition. In practice, surrogate methods are used with gradient descent and tuned with a fixed *small* number of inner loop steps (e.g. [1,2]). Currently, there are no other works or analysis even in the scalar minimization case that explain why a small number of gradient descent should suffice.
> We believe the $\alpha$-descent condition in this regard to be very practical as it does not require full optimization of the inner loop like other approaches (e.g. [1]).

---

> > ### Author Response · Authors · 2024-11-19
> > **Author Response Continued**
> >
> > > Fig 1: It is hard to tell the methods apart, maybe use different line styles.
> >
> > We agree that Figure 1 can be a bit difficult to read.  We have updated the figure with larger text and believe it to more legible. Unfortunately, due the the many methods on the leftmost plot we find that different line styles are difficult to pick out and believe the new figure to be easier to read.
> >
> > References
> >
> > [1] Lavington, J.W., Vaswani, S., Babanezhad Harikandeh, R., Schmidt, M. &amp; Le Roux, N.. (2023). Target-based Surrogates for Stochastic Optimization. <i>Proceedings of the 40th International Conference on Machine Learning</i>, in <i>Proceedings of Machine Learning Research</i> 202:18614-18651 Available from https://proceedings.mlr.press/v202/lavington23a.html.
> >
> > [2] Sharan Vaswani, Olivier Bachem, Simone Totaro, Robert Müller, Shivam Garg, Matthieu Geist, Marlos C. Machado, Pablo Samuel Castro, Nicolas Le Roux Proceedings of The 25th International Conference on Artificial Intelligence and Statistics, PMLR 151:8619-8649, 2022.

---

### Author Response · Authors · 2024-11-19
**Author Response to All Reviewers**

We would like to thank all reviewers for their constructive feedback and comments that will improve the paper. Below we list changes we have made to the paper since the original submission.

* Improved readability of Figure 1 and 2  with larger text and lines
* Figure 3 changed to be averaged over many trajectories including error bars
    * This allows for a more robust comparison of the different methods as opposed to using one trajectory like in [1]
* Added PL definition to the appendix
* Added deep RL experiment with a **16 layer** mlp in mujoco to demonstrate the scalability of our approach. See Figure 10.

References:

[1] Dimitri P Bertsekas. Projected equations, variational inequalities, and temporal difference methods. Lab. for Information and Decision Systems Report LIDS-P-2808, MIT, 2009.

---

### Meta-Review · Area_Chair_Qn8W · 2024-12-20

**Metareview:**

This paper studies a class of variational inequalities by assuming hidden monotonicity. The paper proposes a surrogate-based approach by constructing a surrogate and employ any optimizer to ensure a sufficient decrease condition. The paper provides some convergence analysis under deterministic and stochastic settings. The experiments on done for some toy problems and RL problems. All reviewers agreed the paper has made some nice contributions and recommend an acceptance.

**Additional Comments On Reviewer Discussion:**

The authors have provided detailed rebuttal. Some reviewers have acknowledge that their concerns have been addressed but also mentioned some weakness in terms of strong assumptions.

---

### Decision · Program_Chairs · 2025-01-22

Accept (Poster)